# TimelyGPT: Recurrent Convolutional Transformer for Long Time-series Representation

## Abstract

Pre-trained models (PTMs) have gained prominence in Natural Language Processing and Computer Vision domains. When it comes to time-series PTMs, their development has been limited. Previous research on time-series transformers has mainly been devoted to small-scale tasks, yet these models have not consistently outperformed traditional models. Additionally, the performance of these transformers on large-scale data remains unexplored. These findings raise doubts about Transformer's capabilities to scale up and capture temporal dependencies. In this study, we re-examine time-series transformers and identify the shortcomings of prior studies. Drawing from these insights, we then introduce a pioneering architecture called Timely Generative Pre-trained Transformer (TimelyGPT). This architecture integrates recurrent attention and temporal convolution modules to effectively capture global-local temporal dependencies in long sequences. The relative position embedding with time decay can effectively deal with trend and periodic patterns from time-series. Our experiments show that TimelyGPT excels in modeling continuously monitored biosignal as well as irregularly-sampled time-series data commonly observed in longitudinal electronic health records. This breakthrough suggests a priority shift in time-series deep learning research, moving from small-scale modeling from scratch to large-scale pre-training.

## 1 Introduction

Time-series data mining holds significant importance in healthcare, given its potential to trace patient health trajectories and predict medical outcomes (Ma et al., 2023b; Eldele et al., 2021; Fawaz et al., 2019). In the field of healthcare, there are two primary categories: continuous and irregularly-sampled time-series data. Continuous time-series, such as biosignals, has been extensively studied in various applications including health monitoring (Stirling et al., 2020), disease classification (Phan et al., 2021), and physical activity prediction (Reiss et al., 2019b). Irregularly-sampled time series is commonly found in clinical records, where spontaneous updates are made to an individual patient's health status (Zhang et al., 2022b). The key challenge is to extract meaningful representation from these time-series, especially when there is limited labeled data available. A promising approach to overcome this constraint is to adopt transfer learning (Ma et al., 2023b). Initially, a model is pre-trained on large datasets to capture temporal representation. This pre-trained model (PTM) is then fine-tuned to adapt to the target domain, which often has limited data.

The recent impressive achievements of Transformer PTMs in Natural Language Processing and Computer Vision domains have inspired growing interest in time-series Transformer PTMs. Time-Series Transformer (TST) uses a self-supervised learning through masking strategy to extract contextual dependencies from time-series (Zerveas et al., 2020). Zhang et al. (2023) developed Cross-Reconstruction Transformer (CRT) to model temporal-spectral relations in time-series by dropping and reconstructing certain parts. Additionally, Transformer PTMs have also been applied to traffic (Zhao et al., 2022), tabular time-series (Padhi et al., 2021), and speech data (Liu et al., 2020; 2021).

Transfer learning by pre-training on large time-series data followed by fine-tuning on small labelled datasets provides a promise avenue, yet the adoption of Transformer PTMs remains limited. Existing studies primarily focus on training from scratch on limited labelled data for long-term time series forecasting (LTSF) or classification tasks (Ma et al., 2023a). These studies often introduce tailored architectures and attention modules to extract complex temporal dependencies (Zhou et al., 2021;

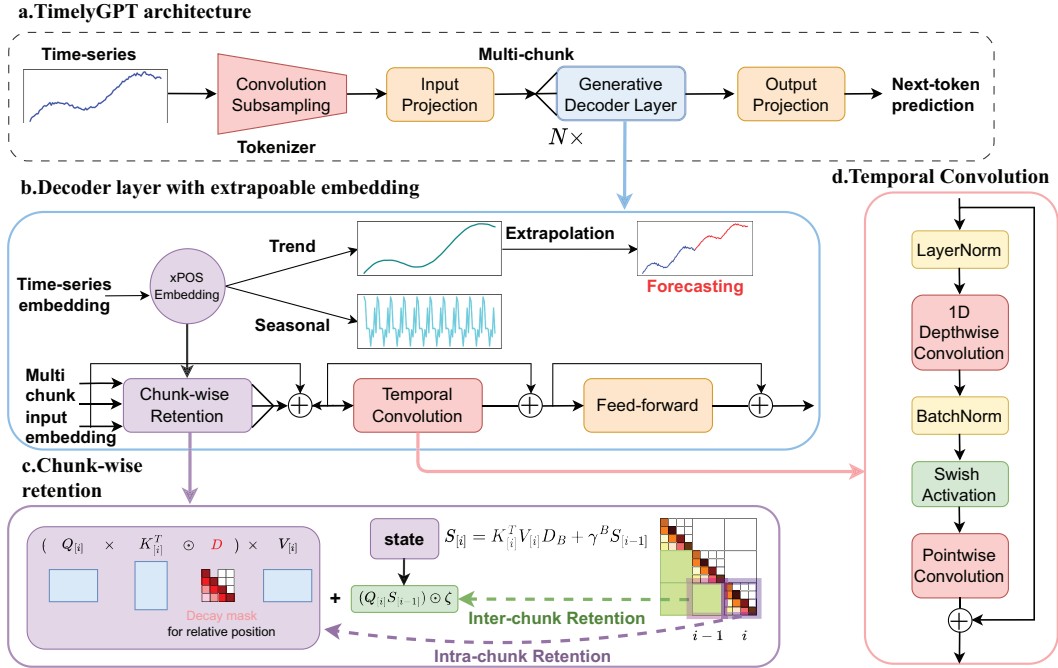

Figure 1: TimelyGPT overview. **a. TimelyGPT architecture** consists of a convolution-subsampling tokenizer followed by a series of decoder layers, with detailed pre-training overflow provided in Appendix B.5. **b.** Each decoder layer is coupled with **extrapolable position embedding** (Section. 3.1) that enables time decomposition and models trends for extrapolation. **c. Chunk-wise Retention**. (Section 3.2) module consists of parallel intra-chunk Retention and recurrent inter-chunk Retention, effectively modeling long sequences. **d. Temporal Convolution** (Section 3.3) captures nuanced local interactions from the temporal representation.

Wu et al., 2022; Zhou et al., 2022). However, the scalability of these transformers on large datasets remains untested (Kaplan et al., 2020). Additionally, recent studies question Transformer's benefits in LTSF, suggesting they may not outperform linear models (Zeng et al., 2022). They argue that permutation-invariant nature of self-attention causes the loss of temporal information. Transformers often underperform compared to convolution-based models in classification, potentially due to their struggles with multi-scale features (Yue et al., 2022; Tang et al., 2022). Overall, existing research on transformers often lacks rigorous evaluation on large datasets and does not consistently outperform conventional approaches on small data. This underscores the pressing need to improve transformer architectures in time-series applications.

In this study, we provide an in-depth analysis of existing time-series transformers, covering key aspects such as the attention mechanism, position embedding, extrapolation, and complexity. We argue that the seemingly inadequacy of the current transformer-based models in modeling time-series data is due to their inability to model long and big time-series data. Once these challenges are resolved, we would observe a typical scaling law comparable to the one observed in the NLP applications. Inspired by our insights, we introduce **Timely Generative Pre-trained Transformer (TimelyGPT)** (Fig. 1) that integrates transformer model with recurrent attention (also known as Retention) and convolution modules. This combination effectively captures both global temporal dependencies and nuanced local interactions in long sequences (Sun et al., 2023; Gulati et al., 2020). TimelyGPT utilizes extrapolatable position embedding (xPOS) to capture trend and periodic patterns in both continuous and irregularly-sampled time-series (Sun et al., 2022). We emphasize the benefits of pre-training on large-scale time-series and then fine-tuning on task-specific small datasets. Experimental results reveal that TimelyGPT can effectively extrapolate temporal representation for ultra-long-term forecasting. Additionally, we found the convolution modules contribute to TimelyGPT's superiority in classification and regression tasks. Together, we advocate for a research shift in the time-series transformers, moving from small-scale modeling from scratch to large-scale pre-training. The key contributions of our research are threefold:

(1) We effectively employ extrapolatable position (xPOS) embedding (Fig. 1b) to extract both trend and periodic patterns in long sequence time-series;

(2) We extend recurrent attention to handle irregularly sampled time-series data (Fig. 1c);

(3) We introduce convolution subsampling tokenization to extract features (Fig. 1a; Fig. 7) from raw time-series input sequence and temporal convolution (Fig. 1d) to sift local temporal interactions among the timesteps.

## 2 REVISITING TRANSFORMER FOR TIME-SERIES

### 2.1 OVERFITTING AND SCALING LAW

Despite the broad applications of transformer-based models in time-series data such as speech (Gulati et al., 2020; Radford et al., 2022), biosignal processing (Song et al., 2023), and traffic flow (Shao et al., 2022; Jiang et al., 2023), their effectiveness in capturing temporal dependencies in LTSF task has been limited and often under-perform compared to linear models Zeng et al. (2022). As Table 1 indicates, time-series transformer models often have more parameters than the size of the training data with only two exceptions, namely large-size Conformer and CRT. Such disparities imply that many forecasting transformers may be over-parameterized, leading to highly variable performance. Our proposed TimelyGPT can effectively pre-train on the large-scale data and follows the scaling law (i.e., positive correlation between the parameter size and the performance) (Section 4.1).

Table 1: The model parameters and utilized datasets of time-series transformers and comparison methods. We consider status as over-parameterization when model parameters $>>$ dataset size. *

| Method | Application | Parameter | Dataset Size (Timesteps) | Param versus Data |
|---|---|---|---|---|
| Informer | Forecasting | 11.3M | 69.7K | Over-param |
| Autoformer | Forecasting | 10.5M | 69.7K | Over-param |
| Fedformer (F/W) | Forecasting | 16.3/114.3M | 69.7K | Over-param |
| PatchTST | Forecasting | 1.2M | 69.7K | Over-param |
| DLinear | Forecasting | 70K | 69.7K | Adequate |
| Conformer (L) | Classification | 118.8M | 55.9B | Adequate |
| CRT | Pre-training | 8.8 M | 109.2M | Adequate |

* The model setup are sourced from papers and default implementation. Details are discussed in Appendix B.1.

### 2.2 SELF-ATTENTION MODULE

As one of the prominent time-series transformers, Conformer utilizes the self-attention mechanism to capture long-range global context in speech data (Gulati et al., 2020). When combined with convolution, Conformer enhances self-attention mechanism by exploiting fine-grained local patterns. Although widely successful, its quadratic complexity has spurred exploration of attention-free modules such as Multi-Layer Perceptron (MLP) (Tolstikhin et al., 2021), implicit long convolution (Poli et al., 2023), and Recurrent Neural Network (RNN) (Peng et al., 2023; Sun et al., 2023). In particular, RNN-based attention modules stand out for their scalability to long sequences with up to 14 billion parameters while maintaining competitive performance and achieving linear training and constant inference complexities. Indeed, the inherent design of RNNs, which has historically proven effective for time-series (Gu et al., 2022), makes them a compelling choice for temporal modeling. However, RNNs struggle with expressiveness Peng et al. (2023). Our TimelyGPT leverages the strengths of both RNNs and transformers for modeling time-series data.

### 2.3 RELATIVE POSITION EMBEDDING

The *absolute* position embedding is inadequate in capturing temporal relations Zeng et al. (2022). While the sinusoidal function utilizes discrete position index, it struggles with extracting positional information from continuous timescale, such as trend and periodic patterns in time-series. In contrast, speech transformers leverage the *relative* position embedding of the T-5 model to handle continuous time (Gulati et al., 2020; Kim et al., 2022), addressing the shortcomings of absolute position embedding(Deihim et al., 2023). The Rotary Position Embedding (RoPE), prevalent in numerous

large language models (Brown et al., 2020; Touvron et al., 2023; Penedo et al., 2023), applies rotation matrices to encode position information from relative distances (Su et al., 2022). Additionally, RNN-based transformer Receptance Weighted Key Value (RWKV) uses exponential decay to integrate time information based on relative positions (Peng et al., 2023). Bridging these techniques, xPOS embedding mitigates oscillations caused by rotary embedding and captures long-term dependencies (Sun et al., 2022). Details can be found in Appendix B.2.

Within healthcare, the increasing susceptibility to illnesses with aging are observed in longitudinal population study using electronic health records (EHR) Ahuja et al. (2022). EHR also exhibit periodic patterns, especially for chronic diseases like COPD with alternating exacerbation and medical treatment effects. Continuous biosignals such as electrocardiogram (ECG) exhibit periodic patterns, reflecting the physiological rhythms of the subject body. We hypotheisze that xPOS can discern trend and seasonal components to capture these patterns in the healthcare time-series data.

## 2.4 EXTRAPOLATING TRANSFORMER ATTENTION WITH XPOS

Existing studies on forecasting transformers have mainly focused on encoder-decoder architectures (Fig. 4a) (Zhou et al., 2021). The input to the decoder is the concatenation of input sequence and a placeholder for the target sequence (i.e., zero padding) of a fixed length. Instead of autoregressively predicting the future outputs, a linear layer is used to output the final output altogether (Zeng et al., 2022). Similarly, encoder-only models leverage encoded embedding for forecasting with the help of a linear layer (Nie et al., 2023) (Fig. 4b). While the poor performance may be due to the limited capacity of the decoder Zhou et al. (2021), recent studies suggest that it may be due to the difficulty of Transformer in representing unseen positions, known as the challenge of length extrapolation (Press et al., 2022). In particular, transformer performance rapidly declines as it starts to forests sequence longer than any of the training sequences (Press et al., 2022). Indeed, both encoder-decoder and encoder-only transformers lack extrapolation ability and rely heavily on their linear layer for forecasting, limiting their effectiveness in LTSF tasks.

The challenge in extrapolation lies in the difficulty of generalizing position embedding to unseen positions. To address this issue, the Attention with Linear Biases (ALiBi) adjusts attention with a penalty linearly correlated with token distance (Press et al., 2022). Building on this, xPOS segments long sequences into multiple chunks and employs exponential decay according to the relative distances (Sun et al., 2022). Consequently, xPOS can handle inference lengths up to eight times the training length, while still maintaining comparable performance. Our TimelyGPT extends xPOS from the NLP domain to time-series forecasting, focusing on exploring the underlying mechanisms that enable the temporal extrapolation.

## 2.5 LONG SEQUENCE WITH QUADRATIC COMPLEXITY

Given the sequence length of $n$, head number of $h$ and embedding size per head of $d$, a full transformer layer has the time complexity of $O(12nh^2d^2+2n^2hd)$ (Detailed derivation in Appendix B.3). Therefore, $2n^2hd > 12nh^2d^2$ if $n > 6hd$. Given a commonly adopted embedding size of $h = 512$, quadratic sequence length $n$ exceeding 3072 becomes the dominant factor. While many existing transformers were designed with efficient attentions, they are only trained on short sequences. Our TimelyGPT is specifically designed to address quadratic complexity with an efficient chunk-wise Retention mechanism for long sequences.

## 3 TIMELYGPT METHODOLOGY

As an overview, our proposed TimelyGPT enables efficient pre-training on unlabeled data via the next-token prediction task to learn the data representation (Fig. 1a). It first processes time-series inputs using a convolution-subsampling tokenizer and input projection for token embedding (Section 3.3). To further extract temporal pattern, TimelyGPT integrates three technical contributions. First, TimelyGPT utilizes xPOS relative position embedding for modeling temporal dependencies to help extrapolation (Fig. 1b, Section 3.1). Second, TimelyGPT utilizes recurrent attention module to capture global content (Fig. 1c, Section 3.2). Third, TimelyGPT deploys convolution module to capture the local content (Fig. 1d, Section 3.3). The latter enables learning the interactions between the global and local content.

## 3.1 RELATIVE POSITION EMBEDDING FOR TEMPORAL PATTERNS

Position embedding is crucial for Transformer, as the self-attention mechanism does not inherently discern token order. Typically, time-series transformer models adopt absolute position embedding that is directly added to the token embedding. In contrast, xPOS computes relative position embedding based on the distance $n - m$ between two tokens $n$ and $m$ (Sun et al., 2022). Given a token embedding matrix $\boldsymbol{X} \in \mathbb{R}^{L \times d}$, xPOS multiplies queries $\boldsymbol{Q}_n = \boldsymbol{X}_n \boldsymbol{W}_Q \in R^{1 \times d_q}$ (and keys $\boldsymbol{K}_m = \boldsymbol{X}_m \boldsymbol{W}_K \in R^{1 \times d_k}$) with position-dependent rotation matrix $e^{i\theta n}$ and exponential decay $\gamma^n$:

$$\tilde{\boldsymbol{Q}}_n = (\gamma e^{i\theta})^n Q_n, \ \tilde{\boldsymbol{K}}_m = (\gamma e^{i\theta})^{-m} K_m, \ \tilde{\boldsymbol{Q}}_n \tilde{\boldsymbol{K}}_m = (\gamma e^{i\theta})^{n-m} Q_n K_m \tag{1}$$

where $\theta$ and $\gamma$ are hyparparameters (Su et al., 2022; Sun et al., 2022). The exponential decay $\gamma^{n-m}$ effectively attenuates the influence of distant tokens, aiding in capturing long-term dependency and extrapolation ability (Sun et al., 2022). Detailed description is available at Appendix B.2.

While originally designed for language modeling, xPOS provides a compelling way for time-series representation, mirroring the seasonal-trend decomposition (Fig. 5). Its exponential decay mechanism naturally focuses its attention on recent data while reducing the influence of older timesteps, reflecting the trend momentum of the series. The incorporation of rotation matrices helps capture the seasonal component of the time-series data through the sinusoidal oscillations. This structure empowers xPOS-based models to represent inherent temporal patterns within time-series data. As our first contribution, TimelyGPT harnesses xPOS's extrapolation capabilities for time-series forecasting, offering insights into the underlying mechanisms driving the temporal trajectories.

## 3.2 RETENTION FOR IRREGULARLY-SAMPLED TIME SERIES

We adapt the Retention mechanism to effectively handle continuous time-series data (Sun et al., 2023). The Retention mechanism based on xPOS can be reformulated as an RNN to naturally model time-series data. Given the queries $\boldsymbol{Q} \in R^{L \times d_q}$ and keys $\boldsymbol{K} \in R^{L \times d_k}$ defined in Eq 1, the forward-pass of the Retention operation can be computed in parallel over all timesteps:

$$\boldsymbol{Q} = (\boldsymbol{X}\boldsymbol{W}_Q) \odot \boldsymbol{R}, \ \boldsymbol{K} = (\boldsymbol{X}\boldsymbol{W}_K) \odot \bar{\boldsymbol{R}}, \ \boldsymbol{V} = \boldsymbol{X}\boldsymbol{W}_V, \ \boldsymbol{R}_n = e^{i\theta n}, \ \bar{\boldsymbol{R}}_m = e^{-i\theta m}$$

$$\text{Retention}(\boldsymbol{X}) = (\boldsymbol{Q}\boldsymbol{K}^\top \odot \boldsymbol{D})\boldsymbol{V}, \ \boldsymbol{D} = \begin{cases} \gamma^{n-m}, & n \geq m \\ 0, & n < m \end{cases} \tag{2}$$

where $\boldsymbol{R} \in R^{L \times d_q}$ is the position-dependent rotation matrix and $\bar{\boldsymbol{R}} \in R^{L \times d_k}$ is the complex conjugate of $\boldsymbol{R}$ (let $d_q = d_k$) (Su et al., 2022). The decay matrix $\boldsymbol{D} \in R^{L \times L}$ captures the temporal information from relative distance between two tokens $n$ and $m$. When reformulated as an RNN, the Retention mechanism is manifested in a recurrent forward-pass with linear time complexity. To handle long sequences, we use chunk-wise Retention by segmenting the sequence into multiple, non-overlapped chunks (Fig. 1c) (Sun et al., 2023). More details are described in Appendix B.4.

As our second contribution, we modify the Retention mechanism to accommodate irregularly-sampled time-series data. Each observation $r_n$ is associated with a specific timestep within a trajectory denoted by $t_{r_n} \in \{1, \dots, T\}$. Here, an irregular time series dataset contains a total of $R$ observations out of $T$ timesteps, represented by $n \in \{1, \dots, R\}$. This non-uniform time intervals pose challenge to traditional time-series methodologies designed for equally spaced data. To handle irregular observations, we modify the decay matrix $\boldsymbol{D}$ in the parallel forward-pass of Retention to accommodate the varying gaps between observations. Given two observations $r_n$ and $r_m$, the decay mask $\boldsymbol{D}$ is adapted according to the time gap $\Delta t$:

$$\text{Retention}(\boldsymbol{X}) = (\boldsymbol{Q}\boldsymbol{K}^\top \odot \boldsymbol{D})\boldsymbol{V}, \quad \boldsymbol{D} = \begin{cases} \gamma^{\Delta t_{m,n}}, & t_{r_n} \geq t_{r_m} \\ 0, & t_{r_n} < t_{r_m} \end{cases}, \quad \Delta t_{m,n} = t_{r_n} - t_{r_m} \tag{3}$$

By taking into account the time interval $\Delta t$ and global state variable $\boldsymbol{S} \in R^{d_k \times d_v}$, we can adapt the recurrent and chunk-wise forward-pass in Eq. 4 and Eq. 5, respectively:

$$\boldsymbol{S}_{r_n} = \gamma^{\Delta t_{n-1,n}} \boldsymbol{S}_{r_{n-1}} + \boldsymbol{K}_{r_n}^\top \boldsymbol{V}_{r_n}, \ \text{Retention}(X_{r_n}) = \boldsymbol{Q}_{r_n} \boldsymbol{S}_{r_n} = \sum_{r_m=1}^{r_n} \gamma^{\Delta t_{m,n}} \boldsymbol{Q}_{r_n} \boldsymbol{K}_{r_m}^\top \boldsymbol{V}_{r_m} \tag{4}$$

$$Q_{[i]} = Q_{B_i:B_{i+1}}, \quad K_{[i]} = K_{B_i:B_{i+1}}, \quad V_{[i]} = V_{B_i:B_{i+1}}, D = \begin{cases} \gamma^{\Delta t_{m,n}}, & t_{r_n} \geq t_{r_m} \\ 0, & t_{r_n} < t_{r_m} \end{cases}$$

$$\text{Retention}(X_{[i]}) = \underbrace{(Q_{[i]}K_{[i]}^\top \odot D)V_{[i]}}_{\text{Intra-chunk}} + \underbrace{(Q_{[i]}S_{[i-1]}) \odot \zeta}_{\text{Inter-chunk}}, \quad \zeta_j = \gamma^j, \text{where } j = r_n$$

$$S_{[i]} = \underbrace{K_{[i]}^\top V_{[i]} \odot D_B}_{\text{Current chunk}} + \underbrace{\gamma^B S_{[i-1]}}_{\text{Past chunk}} \tag{5}$$

where $i$ is the chunk index and $j \in \{1, \ldots, B\}$ is the time index of an observed timestep within a chunk. $D$ is the decay matrix with the chunk size $B$ and $D_B$ is the last row of decay matrix $D$.

### 3.3 CONVOLUTION MODULES FOR LOCAL INTERACTION

Convolution methods excel at identifying localized interactions from time-series (LeCun & Bengio, 1998). They have been explored for extracting multi-scale patterns, characterizing information across varying time scales (Tang et al., 2022). As the first part of our third contribution, we propose using a **convolution-subsampling tokenizer** for feature extraction from the raw time-series input (Fig. 1b). As shown in Fig. 7, it employs multiple 1-D convolution layers to both condense the time dimension and aggregates local information of the input. The convolution-subsampling tokenizer consists of two 1-D convolution with kernel size 3 and stride 2, reducing the sequence length to 1/4. Unlike the prevalent patching technique, which merely segments adjacent timesteps and features (Nie et al., 2023), convolution tokenizer captures local temporal patterns and multi-scale features. More details are described in Appendix B.5.

To sift local temporal interactions in the input sequence (Wu et al., 2020), as the second part of our third contribution, we propose a depth-wise separable convolution as a **temporal convolution module**, consisting of depth-wise and point-wise convolution (Chollet, 2017). As depicted in Fig. 1d, this module starts with a layer normalization, followed by a 1-D depth-wise convolution and a point-wise convolution layers. Batch normalization and swish activation are applied subsequently to depth-wise convolution layer.

## 4 BENCHMARK EXPERIMENT

### 4.1 TRANSFORMER'S SCALABILITY IN TIME-SERIES

We assessed the scalability of TimelyGPT, Informer (Zhou et al., 2021), and DLinear baseline (Zeng et al., 2022) on a large-scale Sleep-EDF dataset (Kemp et al., 2000). We split the 1.2 billion timesteps into 80% timesteps for training, 10% timesteps for validation, and 10% timesteps for testing. Transformers with fixed parameters perform poorly on small-scale data (Fig. 2a), consistent with the finding by Zeng et al. (2022). However, their performance improves as data grow. Moreover, TimelyGPT improves as parameters increase (Fig. 2b), which is attributed to its capacity to handle more data, known as the scaling law for Transformer (Kaplan et al., 2020). Smaller models quickly reach performance plateau with larger dataset size, indicating underfitting (Kaplan et al., 2020). Therefore, transformers are not ideal for small-scale data with limited parameters but are effective as backbone models for large-scale data.

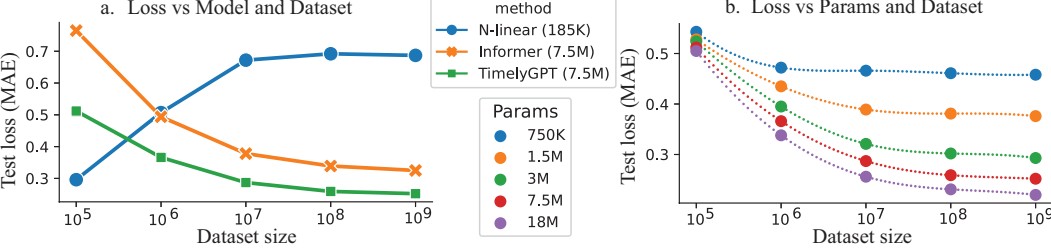

Figure 2: **a.** Forecasting performance of transformers increases with more data. **b.** TimelyGPT with more parameters tends to exhibit better performance when trained on larger datasets.

## 4.2 Ultra-long-term Forecasting and Extrapolation

**Datasets.** We evaluated TimelyGPT on the Sleep-EDF dataset (Kemp et al., 2000) with a sampling rate of 100 Hz, comprising 7 types of biosignals collected from 153 individuals over 20 days, amounting to 1.2 billion timesteps. This dataset size is significantly larger compared to the smaller datasets used in previous studies (Zhou et al., 2021; Wu et al., 2022; Zhou et al., 2022; Nie et al., 2023). The dataset was split into training (80%), validation (10%), and test (10%) sets. All models were pre-trained on the entire training set and fine-tuned on a 20% subset of training data.

**Baselines and Experimental Settings.** We tested TimelyGPT against existing methods including Informer (Zhou et al., 2021), Autoformer (Wu et al., 2022), FEDformer (Zhou et al., 2022), and DLinear (Zeng et al., 2022). Since these transformers lack pre-training strategy, we implemented the self-supervised learning regime with masking (Zerveas et al., 2020). Based on the scaling law in Section 4.1, we set model parameters for all transformers to around 18 million (see the detailed architecture in the Appendix C.5). We avoided redundancy by segmenting time-series into non-overlapping sequences. Since medical monitoring needs ultra-long-term forecasting, we chose input length of 4096 for TimelyGPT pre-training. During fine-tuning, we used look-up window of 2048 as prompt and forecasting windows of 720, 2000, and 6000. We used Mean Absolute Error (MAE) as metrics. Experimental details and baseline setups are in Appendix C.5.

**Results.** In the forecasting experiment, DLinear model was effective for a 720-timestep forecasting window (Fig. 3a). Notably, PatchTST achieved the best MAE at 0.456, whereas TimelyGPT confers comparable performance for shorter sequences. As the forecasting window increases to 2000 and 6000 timesteps, DLinear model suffered a performance drop due to the limited model parameters. The declined performance of the transformer baselines is attributed to their reliance on the linear layers without decoding future times. TimelyGPT maintained consistently superior extrapolation performance even in ultra-long-term forecasting (6000 timesteps).

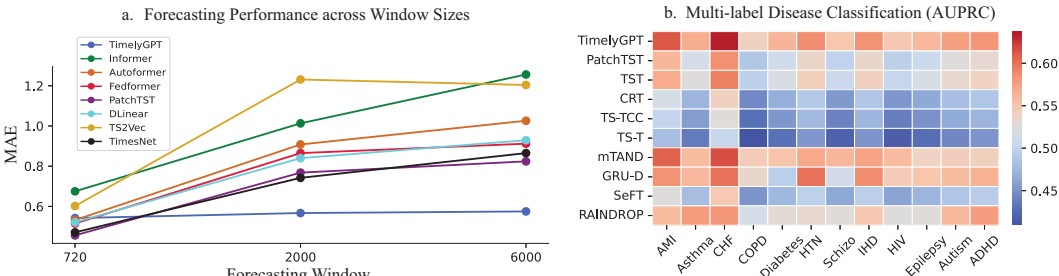

Figure 3: **a.** MAE for 8 methods in forecasting biosignals from the Sleep-EDF dataset with 3 forecasting window sizes. **b.** Phenotype prediction using the PopHR data. The heatmap displays the AUPRC for the 9 baselines (rows) in classifying the 12 phenotypes (columns). Full forecasting and classification results can be found in Table 9 and Table 10, respectively.

## 4.3 Classification and Regression

**Datasets.** For classification tasks, we pre-trained on the EEG Fpz-Cz from the Sleep-EDF dataset (sleep stage) (Kemp et al., 2000) and the PTB-XL dataset (Alday et al., 2020) separately. Subsequently, the PTMs were fine-tuned on the Sleep-EDF, Epilepsy (Andrzejak et al., 2001), PTB-XL, and EMG (Goldberger et al., 2000) datasets. For regression tasks, we pre-trained on both the PTB-XL and PPGDalia datasets (Reiss et al., 2019a), followed by fine-tuning on the IEEEPPG (Zhang et al., 2015), RR, HR, and SpO2 (Tan et al., 2021) datasets. Each dataset was split into a training set (80%), validation set (10%), and test set (10%). When both pre-training and fine-tuning on the same dataset, we fine-tuned the PTMs using 20% of the training set.

**Baselines and Experiment Settings.** We assessed various transformers, including PatchTST (Nie et al., 2023), AutoFormer (Wu et al., 2022), and FedFormer (Zhou et al., 2022). We also evaluated Transformer PTMs including TST (Zerveas et al., 2020), TimesNet (Wu et al., 2023), and CRT (Zhang et al., 2023) as well as consistency-based PTMs including TS-TCC (Eldele et al., 2021), TS2Vec Yue et al. (2022), and TF-C (Zhang et al., 2022c). Consistency-based PTMs aim to minimize distance of representation from the positive pairs and maximize the distance of representation from

Table 2: Comparing TimelyGPT with baseline models on the downstream classification and regression tasks. Bold-faced font are the best value in each task among the 11 methods.

| Task | Classification (Accuracy %) | | | | Regression (MAE) | | | |
|---|---|---|---|---|---|---|---|---|
| Pre-training | Sleep-EDF | | PTB-XL | | PTB-XL & PPGDalia | | | |
| Fine-tuning | Sleep-EDF | Epilepsy | PTB-XL | EMG | IEEEPPG | RR | HR | SpO2 |
| TimelyGPT | 89.21 | **92.79** | 86.52 | **95.87** | 26.17 | **2.78** | **8.53** | **4.26** |
| PatchTST | 89.57 | 91.27 | 83.42 | 95.23 | **26.08** | 2.89 | 9.46 | 4.45 |
| AutoFormer | 78.86 | 84.21 | 78.51 | 88.56 | 32.18 | 4.13 | 13.29 | 4.95 |
| FedFormer | 76.43 | 81.65 | 75.53 | 85.21 | 31.11 | 4.36 | 13.82 | 4.75 |
| TimesNet | 83.58 | 85.96 | 79.38 | 89.26 | 29.95 | 4.19 | 13.65 | 4.83 |
| TST | 88.83 | 88.02 | 81.86 | 94.16 | 26.81 | 3.47 | 12.63 | 4.95 |
| CRT | **90.12** | 91.05 | **87.81** | 94.56 | 26.52 | 2.96 | 9.02 | 4.48 |
| TS2Vec | 86.21 | 88.27 | 82.65 | 93.77 | 27.89 | 3.53 | 11.56 | 4.60 |
| TS-TCC | 86.06 | 89.74 | 84.66 | 93.25 | 29.32 | 4.09 | 13.64 | 4.86 |
| TF-C | 86.56 | 87.52 | 82.71 | 93.83 | 28.52 | 4.38 | 14.15 | 4.87 |
| LSTM | 80.15 | 76.51 | 78.83 | 86.95 | 30.15 | 4.95 | 14.37 | 5.05 |

the negative pairs, learning effective representation for downstream tasks (Ma et al., 2023b). We used accuracy and MAE as evaluation metrics for classification and regression, respectively.

**Results.** TimelyGPT achieved the best performance in classifying Epilepsy and EMG and in the regression of IEEEPPG, HR, and SpO2 (Table 2). This superiority highlights the potential of generative pre-training framework to generalize across datasets (Radford et al., 2019). On the other hand, CRT stood out the best when both pre-training and fine-tuning were conducted on the same dataset. When dealing with the short sequences, particularly in the Epilepsy, EMG, and IEEEPPG datasets, the performances of PatchTST and TST are close. PatchTST surpassed TST in modeling long sequences in the Sleep-EDF dataset, hinting at the benefits of its patching tokenizer. As TimelyGPT outperformed PatchTST, we reasoned that our proposed convolution-subsampling tokenizer might be more adequate at extracting dense information from long sequences. The masking pre-training strategy introduces distribution shift (Zhang et al., 2023) that adversely affects Autoformer and Fedformer, which rely on time decomposition and frequency domain information, respectively. This pre-training strategy disrupt specific data characteristics, leading to underperformance in these models compared to other transformers. We also observed the underperformance of TimesNet, which heavily depends on frequency-domain information and Fourier transform. Furthermore, consistency-based PTMs generally exhibited lower performance in these tasks.

## 4.4 Classification on Irregularly-Sampled Time Series

**PopHR Dataset and pre-processing.** The Population Health Record (PopHR) was established to monitor population health in Montreal, Quebec, Canada (Shaban-Nejad et al., 2016; Yuan et al., 2018). The administrative data consists of the medical histories of 1.2 million patients in the form of International Classification of Diseases (ICD) codes. These are irregularly-sampled time-series data due to the spontaneous out-patient hospital visits. We used the PheWAS catalog to map ICD-9 codes to phenotype codes (PheCodes) (Denny et al., 2013; 2010) (Appendix C.7). PopHR has a set of predefined rule-based labels for 12 phenotypes, allowing evaluating multi-label classification.

**Experiment Design and baseline.** We used 47,000 patients with at least 10 codes. The datasets were split into training (80%), validation (10%), and test (10%) sets. We pre-trained on the entire training set and fine-tuned on a 20% subset of training data. For comparative analysis, we compared TimelyGPT to PTMs, which have demonstrated effective performances in Section 4.3, including PatchTST, TST, CRT, TS-TCC, and TF-C. Moreover, we evaluated existing methods that account for irregular time-series data, including mTAND (Shukla & Marlin, 2021), GRU-D (Che et al., 2018), SeFT (Horn et al., 2020), and RAINDROP (Zhang et al., 2022a). We used cross entropy and Area under Precision Recall Curve (AUPRC) to evaluate pre-training and fine-tuning, respectively.

**Results.** TimelyGPT exceeds other PTMs for multi-label classification (Fig. 3b). In contrast to encoder-only TST and PatchTST, TimelyGPT effectively captured trend patterns and handled unequal time intervals. This finding aligns with the limitations of absolute position embedding in

the processing of irregular sequences with phase shifting issues (Sinha et al., 2022). Furthermore, the presence of irregular intervals introduces noise in the frequency domain, impairing the effectiveness of CRT, TS-TCC, and TF-C approaches. Compared to methods designed for irregularly sampled time-series, TimelyGPT demonstrated superior performance (average AUPRC of 57.6%) with the runner-up being mTAND (56.4%).

## 4.5 ABLATION STUDY

To assess the contributions of various components in the TimelyGPT model, we conducted ablation studies by selectively **omitting** components including convolution subsampling tokenizer, temporal convolution module, exponential decay, and the relative position embedding RoPE. Notably, removing all components results in a vanilla GPT-2. Since exponential decay in xPOS depends on RoPE, we cannot assess the impact of exponential decay independently by removing the RoPE component. Our ablation study consisted of two classification experiments on Sleep-EDF and PopHR's irregularly-sampled time-series (CHF phenotype only). The models were pre-trained and then fine-tuned on the same dataset. The fine-tuning results are available at Appendix C.8.

In the Sleep-EDF experiment, RoPE was the main contributor to the improvement over the baseline GPT-2, followed by exponential decay. Relative position embedding in TimelyGPT was found to effectively capture temporal dependencies. Moreover, by integrating convolution modules, TimelyGPT was able to capture local interaction, achieving close performance to other transformers. However, the convolution modules had limited benefits for irregular-sampled time-series, possibly due to its discrete nature. By mitigating the effects of distant diagnoses, exponential decay extracted trend patterns in patients' health trajectory, making it promising for irregular time-series analysis. Pre-training conferred a notable increase of 3.89% in biosignal classification, while the improvement was less evident in irregularly-sampled time series (0.83%).

## 4.6 VISUALIZATION ANALYSIS OF THE FORECAST TRAJECTORIES

We qualitatively examined the ultra-long-term forecasting for TimelyGPT alongside the best baseline PatchTST, DLinear, and ablating methods, focusing on sleep stage transitions (Appendix C.10). We include a look-up window (i.e., prompt) and a 6000-timestep forecasting window. Forecasting beyond 4000 timesteps was marked as extrapolation because it exceeded the training length.

In the rectal temperature (i.e., trend signal), TimelyGPT generates forecasts that align well with the groundtruth (Fig. 10). Notably, the small bump in the prompt right before the 1000-th timestep is a typical indicator for temperature drop. Most models are able to capture that except for DLinear. Beyond the training length of 4000, TimelyGPT continues to excels by accurately predicting the rise of the rectal temperature at around 7000-th timestep whereas PatchTST and GPT-2 failed to capture that. We also visualized foresting EEG periodic biosignal and found similar conclusion (Fig. 9). In contrast, both PatchTST and vanilla GPT-2 experience a decline in performance, suggesting a dependency onn linear mapping as discussed in Section. 2.4 and in previous research (Li et al., 2023). TimlyGPT exhibits superior extrapolation capabilities over the ablating baseline GPT-2 with RoPE, highlighting its effective trend pattern modeling for extrapolation.

## 5 CONCLUSION AND FUTURE WORK

TimelyGPT is designed to model long sequence time-series by incorporating xPOS embedding, recurrent attention, and convolution modules. Our experiments show that TimelyGPT confers accurate ultra-long-term forecast up to 6000 timesteps through its extrapolation capabilities. Moreover, TimelyGPT can also effectively capture temporal patterns from irregularly-sampled time-series data via the time decay mechanism.

The current design of xPOS is confined to unidirectional attention (Press et al., 2022), which restricts its application as Transformer encoder. Future work will aim to address these technical challenges to enhance adaptability and extrapolation. Transferring between domains can further improve the performance across various biosignals. Additionally, we emphasize the need to prevent misuse of pre-trained models in forecasting. It can lead to serious privacy risks, such as unauthorized prediction of patient health status and breaches of privacy.

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

# A  RELATED WORKS

## A.1  REVIEW OF EXISTING WORKS

**Efficient Attention in Transformer:** Transformer models have found extensive applications in both the Natural Language Processing and Computer Vision domains (Vaswani et al., 2023). In the standard softmax self-attention (Vaswani et al., 2023), the $i$-th token for queries, keys, and values are denoted as $Q_i \in R^{N \times d}$, $K_i \in R^{N \times d}$, and $V_i \in R^{N \times d}$ respectively, where each token index $i$ ranges from 1 to $N$. The output embedding for the $i$-th token is represented as $\boldsymbol{O}_i = \frac{\sum_j^N \text{sim}(\boldsymbol{Q}_i, \boldsymbol{K}_j)\boldsymbol{V}_j}{\sum_{j=1}^N \text{sim}(\boldsymbol{Q}_i, \boldsymbol{K}_j)}$, where the the similarity function represents the softmax of inner-product $\text{sim}(\boldsymbol{Q}_i, \boldsymbol{K}_j) = \exp(\boldsymbol{Q}_i \boldsymbol{K}_j^\top / \sqrt{d})$. The self-attention mechanism, also known as token-mixer, aims to integrate information from every token and thus capture global-range interaction. However, computing the dot product $\boldsymbol{Q}_i \boldsymbol{K}_j^\top$ before the softmax operation introduces computational complexity of $O(N^2 d)$. As sequence length increases, this quadratic complexity becomes bottleneck, making transformers challenging to train for longer sequences. Numerous studies have been proposed to address the quadratic issue in self-attention mechanism. Notably, linear attention replaces the softmax term $\text{sim}(\boldsymbol{Q}_i, \boldsymbol{K}_j)$ with $\phi(\boldsymbol{Q}_i)\phi(\boldsymbol{K}_j^\top)$ for a nonlinear kernel function $\phi(\cdot)$ (Katharopoulos et al., 2020), avoiding the quadratic computation.

Recent research has explored alternatives to the token-mixer attention mechanism including Multi-Layer Perceptron (MLP) (Tolstikhin et al., 2021), convolution (Poli et al., 2023), and RNN (Peng et al., 2023; Sun et al., 2023). Particularly, RNN-variant models like RWKV and RetNet have successfully scaled up to more than 14 billion parameters, yielding comparable performance to conventional transformers. A fascinating connection between linear attention and RNNs has been identified (Katharopoulos et al., 2020), making RNN-based token mixer as efficient as linear attention. The output embedding from linear attention can be recast as a RNN: $\boldsymbol{O}_i = \frac{\phi(\boldsymbol{Q}_i) \sum_j^N \phi(\boldsymbol{K}_j^\top) \boldsymbol{V}_j}{\phi(\boldsymbol{Q}_i) \sum_j^N \phi(\boldsymbol{K}_j^\top)} = \frac{\phi(\boldsymbol{Q}_i)\boldsymbol{S}_i}{\phi(\boldsymbol{Q}_i)\boldsymbol{Z}_i}$, where $\boldsymbol{S}_i = \sum_j^N \phi(\boldsymbol{K}_j^\top)\boldsymbol{V}_j$, $\boldsymbol{Z}_i = \sum_j^N \phi(\boldsymbol{K}_j^\top)$. The output embedding $\boldsymbol{O}_i$ then depends on both $\boldsymbol{S}_i$ and $\boldsymbol{Z}_i$, which are incrementally updated through cumulative sums. Thus, the RNN-based token-mixer not only competes in performance, but also offers linear training and consistent inference complexities. By employing exponential decay mechanism, it diminishes the influence of distant positions, transitioning from "token-mixing" to "time-mixing". Considering RNN's historical effectiveness in time-series and audio domains, it stands out as an excellent choice for temporal modeling.

**State Space Model:** Recent advancements in deep state-space models (SSMs) have led to significant improvements in long sequence modeling across areas including Natural Language Processing, Computer Vision, audio, and time-series (Gu et al., 2022). These models learn continuous or discrete-time representation, transforming input signals into output via state variables. Mainstream sequence models face memory forgetting issues, such as the fixed-size context windows of Transformer and the vanishing gradient in RNNs. In response, HiPPO presents a closed-form solution adept at preserving historical information efficiently. Its variant HiPPO-LegS not only addresses vanishing gradients but also provides timescale robustness, ensuring resilience to different sampling intervals. Furthermore, the Hungry Hungry Hippos (H3) model exhibits capabilities to recall early tokens and draw information throughout a sequence. However, SSMs often fail to leverage content-aware variables as Transformer. In this context, the RetNet model can be likened to the Structured State Space Sequence (S4) model (Gu et al., 2022), especially when queries and keys are not dependent on data.

**Time-series Transformer:** Transformers are increasingly applied in LTSF tasks, attributed to their capabilities in capturing long-term temporal dependencies (Zhou et al., 2021; Wu et al., 2022; Zhou et al., 2022; Woo et al., 2022; Nie et al., 2023). Researchers have modified transformers by incorporating custom attention modules to address complex temporal dependencies (Zhou et al., 2021; Wu et al., 2022; Zhou et al., 2022). Studies like (Wu et al., 2022; Zhou et al., 2022; Woo et al., 2022) have introduced time-decomposition techniques into attention mechanisms to bolster modeling capability. The majority of studies focus on the encoder-decoder architecture, coupled with a one-forward prediction framework (Zhou et al., 2021). In this design, the decoder takes a concatenated input of the context (or prompt) and placeholder forecasting windows, directly generating the resulting embedding without autoregressive decoding. As a result, these models aim to avoid error

accumulation seen in autoregressive frameworks, but aligning its performance closely with linear models (Zeng et al., 2022). Encoder-only models, like patchTST, use the encoded embedding for forecasting with the help of a linear layer (Nie et al., 2023). Additionally, self-supervised representation learning techniques in time series, such as TS2Vec and TimesNet, offer valuable representation learning capabilities for forecasting tasks (Yue et al., 2022; Wu et al., 2023).

**Challenges for Time-series Transformer Models:** Recent studies show that transformers are limited in time-series forecasting (Zeng et al., 2022). The permutation-invariant nature of self-attention and the limitations of position embedding have been cited as potential issues, leading some to argue that transformers may not significantly outperform traditional models in this domain.

## A.2 FORECASTING TRANSFORMER MODELS

In the forecasting experiment, we conducted a comparison between our proposed TimelyGPT and other models such as encoder-decoder transformers (such as Informer, Autoformer, and Fedformer) and encoder-only transformer (PatchTST). The encoder-decoder architecture utilizes one-forward decoding instead of autoregressive decoding, which is hard to generate effective representation of future timesteps. The PatchTST model exclusively focuses on encoding embedding of input sequence. Both encoder-decoder and encoder-only architectures lack the ability to generate representative embedding of future time and instead rely on linear layers for forecasting. It is important to note that both encoder-decoder and encoder-only transformers lack the ability to extrapolate due to the presence of absolute position embedding.

Table 3: Properties of transformer for time-series forecasting.

| Method | Architecture | Decoding | Rely on Linear | Extrapolation |
|---|---|---|---|---|
| TimelyGPT | Decoder-only | Autoregressive | No | Yes |
| Informer | Encoder-decoder | One-forward | Yes | No |
| Autoformer | Encoder-decoder | One-forward | Yes | No |
| Fedformer | Encoder-decoder | One-forward | Yes | No |
| PatchTST | Encoder-only | — | Yes | No |

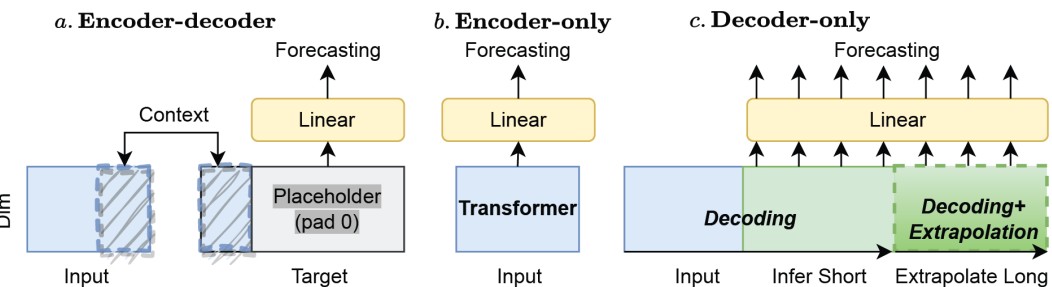

Figure 4: **Forecasting transformer architectures**. **a**. The encoder-decoder architecture employs a one-forward prediction without autoregressive decoding for forecasting. **b**. The encoder-only architecture directly projects the input representation to make forecasting. **c**. The proposed decoder-only architecture in TimelyGPT decodes embedding for future timesteps autoregressively and utilizes extrapolation technique for foresting unseens timesteps for sequence longer than any of training sequences.

## A.3 PRE-TRAINED MODELS FOR CLASSIFICATION AND REGRESSION

For the time-series classification and regression tasks, we compared our proposed TimelyGPT with forecasting transformers (such as Autoformer, Fedformer, and PatchTST), transformer-based PTMs (TST and CRT), and consistency-based PTMs (TS-TCC and TF-C). Forecast transformers utilize the pre-training strategy with the mask introduced by (Zerveas et al., 2020). Additionally, both TST and

CRT utilize pre-training strategies that involve masking and droping task. The consistency-based PTMs, TS-TCC and TF-C, propose contextual contrasting modules that utilize the contexts from the temporal contrasting module.

Table 4: Properties of transformer for time-series forecasting.

| Method | Architecture | Pre-training Type |
|---|---|---|
| TimelyGPT | Decoder-only | Transformer-based |
| Autoformer | Encoder-decoder | Transformer-based |
| Fedformer | Encoder-decoder | Transformer-based |
| PatchTST | Encoder-only | Transformer-based |
| TST | Encoder-only | Transformer-based |
| CRT | Encoder-decoder | Transformer-based |
| TS-TCC | Contrastive Learning | Consistency-based |
| TF-C | Contrastive Learning | Consistency-based |

# B DETAILS ABOUT TRANSFORMER ARCHITECTURE AND TIMELYGPT METHODOLOGY

## B.1 MODEL HYPERPARAMETERS AND OVERALL SETUP

**Time-series Transformers and DLinear baseline**: For the time-series transformers and the DLinear model, we set the both look-up and forecasting windows as 96. The model parameters of transformers are invariant to the sequence length. However, Fedformer introduces Fourier modules that perform a linear transformations in the frequency domain, resulting in additional parameters with respect to sequence length. Furthermorer, both PatchTST and DLinear models utilize a fully-connected linear layer for prediction, which is dependent on the window size.

**Conformer:** a speech transformer model evaluated on the large-scale LibriSpeech dataset (Panayotov et al., 2015). This dataset consists of 970 hours of labeled speech and a text-only corpus with 800 million word token. Conformer extracted 80-channel filterbanks features computed from a 25ms window with a stride of 10ms. Hence, the 970-hour speech data can be processed as 349.2 million sequences. The Conformer model has three hyperparameters for small, medium, and large models. The paper provides the total number of parameters.

**CRT**: a transformer-based PTMs especially designed for mid-scale time-series. Based on the default hyperparameter setup in implementation (`https://github.com/BobZwr/Cross-Reconstruction-Transformer`), the total number of parameters in the model is 3.2 million. It includes 6 encoder layer, 2 decoder layer, 8 heads, and 128 hidden dimension for both the model and feed-forward layer. Additional parameters are set with default values.

Table 5: The model parameters and experimental setup of time-series transformers are contrasted with comparison methods.

| Method | Application | Model Dim | Enc | Dec | Parameter | Dataset Size |
|---|---|---|---|---|---|---|
| Informer | Forecasting | 512 | 2 | 1 | 11.3M | 69.7K |
| Autoformer | Forecasting | 512 | 2 | 1 | 10.5M | 69.7K |
| Fedformer (F/W) | Forecasting | 512 | 2 | 1 | 16.3/114.3M | 69.7K |
| PatchTST | Forecasting | 128 | 3 | — | 9.5M | 69.7K |
| DLinear | Forecasting | — | 1 | — | 70K | 69.7K |
| Conformer (L) | Classification | 512 | 17 | 1 | 118.8M | 55.9B |
| CRT | Pre-training | 128 | 6 | 2 | 8.8 M | 109.2M |

### B.2 FROM ABSOLUTE TO RELATIVE POSITION EMBEDDING

Unlike RNNs or CNNs, the inclusion of positional embedding is essential for the Transformer model. Since the permutation-invariant self-attention mechanism cannot capture input order, making it challenging to differentiate tokens in various positions. The solution fall into two categories: (1) incorporate position information into the inputs, i.e., absolute position embedding; (2) modify the attention matrix to distinguish tokens at different positions, referring to relative position embedding.

In absolute position embedding, the token representation for a given token $n$ consists of a word embedding $\boldsymbol{X}_n$ and a position embedding $\boldsymbol{P}_n$. The self-attention mechanism is expressed as:

$$\boldsymbol{Q}_n = (\boldsymbol{X}_n + \boldsymbol{P}_n)\boldsymbol{W}_Q, \quad \boldsymbol{K}_n = (\boldsymbol{X}_n + \boldsymbol{P}_n)\boldsymbol{W}_K, \quad \boldsymbol{V}_n = (\boldsymbol{X}_n + \boldsymbol{P}_n)\boldsymbol{W}_V$$

$$\boldsymbol{A}_{n,m} = \text{softmax}(\boldsymbol{Q}_n\boldsymbol{K}_m^\top), \quad \boldsymbol{O}_m = \sum_j \boldsymbol{A}_{n,m}\boldsymbol{V}_m \tag{6}$$

where $\boldsymbol{A}_{n,m}$ is an attention score between token $n$ and $m$ without scaling. The inner-dot product $\boldsymbol{Q}_n\boldsymbol{K}_m^\top$ and output embedding $\boldsymbol{O}_m$ can be expanded as follows:

$$\boldsymbol{Q}_n\boldsymbol{K}_m^\top = (\boldsymbol{X}_n + \boldsymbol{P}_n)\boldsymbol{W}_Q((\boldsymbol{X}_m + \boldsymbol{P}_m)\boldsymbol{W}_K)^\top = (\boldsymbol{X}_n + \boldsymbol{P}_n)\boldsymbol{W}_Q\boldsymbol{W}_K^\top(\boldsymbol{X}_m + \boldsymbol{P}_m)^\top$$

$$= \underbrace{\boldsymbol{X}_n\boldsymbol{W}_Q\boldsymbol{W}_K^\top\boldsymbol{X}_m^\top}_{\text{token-token}} + \underbrace{\boldsymbol{X}_n\boldsymbol{W}_Q\boldsymbol{W}_K^\top\boldsymbol{P}_m^\top}_{\text{token-position}} + \underbrace{\boldsymbol{P}_n\boldsymbol{W}_Q\boldsymbol{W}_K^\top\boldsymbol{X}_m^\top}_{\text{position-token}} + \underbrace{\boldsymbol{P}_n\boldsymbol{W}_Q\boldsymbol{W}_K^\top\boldsymbol{P}_m^\top}_{\text{position-position}} \tag{7}$$

$$\boldsymbol{O}_n = \sum_m \text{softmax}\left((\boldsymbol{X}_n\boldsymbol{W}_Q + \boldsymbol{P}_n\boldsymbol{W}_Q)(\boldsymbol{W}_K^\top\boldsymbol{X}_m^\top + \boldsymbol{W}_K^\top\boldsymbol{P}_m^\top)\right)(\boldsymbol{X}_n + \boldsymbol{P}_m)\boldsymbol{W}_V \tag{8}$$

where attention arises from four types of interactions: (1) token-token interaction; (2) token-position interaction; (3) position-token interaction; (4) position-position interaction. However, absolute position embedding only incorporates fixed position information, neglecting the relative positional difference between the token $n$ and $m$.

In the realm of audio processing, prevalent transformers like Conformer (Gulati et al., 2020) incorporate relative positional information through the T5 position embedding (Raffel et al., 2020). Notably, the T5 model suggests a minimal interaction between tokens and positions, resulting in the exclusion of token-position and position-token terms from the attention matrix:

$$\boldsymbol{Q}_n\boldsymbol{K}_m^\top = \boldsymbol{X}_n\boldsymbol{W}_Q\boldsymbol{W}_K^\top\boldsymbol{X}_m^\top + \boldsymbol{X}_n\boldsymbol{W}_Q\boldsymbol{W}_K^\top\boldsymbol{P}_m^\top + \boldsymbol{P}_n\boldsymbol{W}_Q\boldsymbol{W}_K^\top\boldsymbol{X}_m^\top + \boldsymbol{P}_n\boldsymbol{W}_Q\boldsymbol{W}_K^\top\boldsymbol{P}_m^\top$$

$$= \boldsymbol{X}_n\boldsymbol{W}_Q\boldsymbol{W}_K^\top\boldsymbol{X}_m^\top + {\color{red}\beta_{n,m}} \tag{9}$$

where the position-position interaction term, $\boldsymbol{P}_n\boldsymbol{W}_Q\boldsymbol{W}_K^\top\boldsymbol{P}_m^\top$, is replaced with a trainable bias related to the position $n$ and $m$. The T5 position embedding follows Transformer-XL, omitting the position term $\boldsymbol{P}_m\boldsymbol{W}_V$ in the attentive aggregation computation (Dai et al., 2019; Yang et al., 2020). As a result, the relative position embedding is only added to the dot product $\boldsymbol{Q}\boldsymbol{K}^\top$:

$$\boldsymbol{O}_n = \sum_m \text{softmax}(\boldsymbol{X}_n\boldsymbol{W}_Q\boldsymbol{W}_K^\top\boldsymbol{X}_m^\top + {\color{red}\beta_{n,m}}){\color{red}\boldsymbol{X}_m\boldsymbol{W}_V} \tag{10}$$

The RoPE technique leverages the property of rotation matrix to model positional information (Su et al., 2022). To incorporate this relative position information into the queries $\boldsymbol{Q}$ and keys $\boldsymbol{K}$, the method aims to identify functions $f_{\boldsymbol{Q}}(\boldsymbol{Q}, \cdot)$ and $f_K(\boldsymbol{K}, \cdot)$ that satisfies following criteria:

$$\langle \boldsymbol{Q}_n, \boldsymbol{K}_m \rangle = \langle f_{\boldsymbol{Q}}(\boldsymbol{Q}, n), f_K(\boldsymbol{K}, m) \rangle = g(\boldsymbol{Q}, \boldsymbol{K}, m - n), \tag{11}$$

where $g$ is a function that depends only on the relative distance $m - n$ and $\boldsymbol{Q} = \boldsymbol{X}\boldsymbol{W}_Q$ and $\boldsymbol{K} = \boldsymbol{X}\boldsymbol{W}_K$ stand for token embedding for queries and keys matrices, respectively. RoPE defines the function $f$ involving a $d$-dimensional rotation matrix $\boldsymbol{R}$:

$$f_{\boldsymbol{Q}}(\boldsymbol{Q}, n) = \boldsymbol{R}_{\Theta,n}^d(\boldsymbol{X}_n\boldsymbol{W}_Q), \quad f_{\boldsymbol{K}}(\boldsymbol{K}, m) = \boldsymbol{R}_{\Theta,m}^d(\boldsymbol{X}_m\boldsymbol{W}_K) \tag{12}$$

With a given hidden size $d$, a block diagonal matrix $\boldsymbol{R}_{\Theta,n}^d$ contains multiple rotation matrices $(\boldsymbol{R}_{n,\theta_1}^{(1)}, \ldots, \boldsymbol{R}_{n,\theta_{d/2}}^{(d/2)})$ on its diagonal:

$$\boldsymbol{R}_{\Theta,n}^d = \begin{bmatrix} \boldsymbol{R}_{n,\theta_1}^{(1)} & & \\ & \ddots & \\ & & \boldsymbol{R}_{n,\theta_{d/2}}^{(d/2)} \end{bmatrix}, \quad \boldsymbol{R}_{n,\theta_i}^{(i)} = \begin{bmatrix} \cos n\theta_i & -\sin n\theta_i \\ \sin n\theta_i & \cos n\theta_i \end{bmatrix}, \quad \theta_i = 10000^{-2(i-1)/d}.$$

$$\tag{13}$$

In RoPE, any even-dimensional representation can be built by placing multiple 2-dimensional rotation matrices diagonally within the $\boldsymbol{R}_{\Theta,n}^d$ matrix, expanding hidden size from 2-dimension to $d$-dimension. As $\boldsymbol{R}_{\Theta,m-n}^d = (\boldsymbol{R}_{\Theta,n}^d)^\top \boldsymbol{R}_{\Theta,m}^d$, RoPE satisfies the property outlined in Eq 11:

$$\langle \boldsymbol{Q}_n, \boldsymbol{K}_m \rangle = \sum_{i=1}^{d/2} \langle \boldsymbol{Q}_n[2i-1:2i], \boldsymbol{K}_m[2i-1:2i] \rangle \tag{14}$$

$$= \sum_{i=1}^{d/2} \boldsymbol{R}_{\theta_i,m-n}^d \langle (\boldsymbol{X}_n \boldsymbol{W}_Q)[2i-1:2i], (\boldsymbol{X}_m \boldsymbol{W}_K)[2i-1:2i] \rangle . \tag{15}$$

In the RoPE approach, relative position information is added to the inner product $\boldsymbol{Q}\boldsymbol{K}^\top$ by rotating the angles of queries and keys matrices. Recently, Sun et al. (2022) argues that the sinusoids used in the rotation matrices do not change monotonically. Instead, they oscillate dramatically as the relative distance increases. This limitation hinders RoPE's ability to sequences of extended lengths. To address it, Sun et al. (2022) proposes xPOS embedding that preserves the advantage of ROPE and behaves stably at long-term dependency by measuring position monotonicity (Sun et al., 2022).

### B.2.1 XPOS EMBEDDING FOR TIME-SERIES

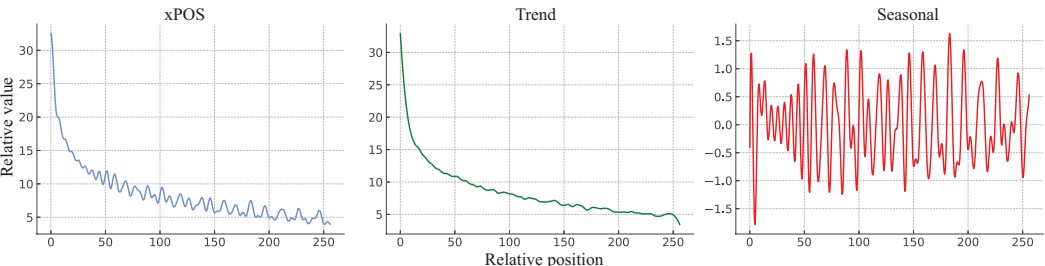

Figure 5: The xPOS embedding diminishes distant temporal information according to the relative distance, enabling decomposition to capture both trend and periodic dynamics in time-series data.

### B.3 TIME COMPLEXITY OF SELF-ATTENTION MODULE

Given a sequence length $n$, head number $h$, and hidden size per head $d$, the hidden dimension of model is $hd$. The self-attention mechanism involves the following procedures: To project input sequence into matrices $Q, K, V$, the input sequence matrix of size $n \times hd$ is multiplied by a projection matrix of size $hd \times hd$. For each head, the matrix $Q$ is multiplied by the transpose of $K$ matrix to compute a squared attention matrix $a$ of size $n \times n$. This attention matrix is then multiplied by the matrix $V$. The resulting matrix goes through a final output projection. Excluding the softmax operator and attention scaling, the computation of the self-attention mechanism can be broken down as follows:

**Projection of** $Q, K, V$**:** $3 \times (n \times hd \times hd) = 3n(hd)^2$

**Attention of all heads:** $h \times (n \times d \times n + n \times n \times d) = h \times (n^2 d + n^2 d) = h(2n^2 d)$

**Output Projection:** $n \times hd \times hd = n(hd)^2$

**Full self-attention:** $3n(hd)^2 + h(2n^2 d) + n(hd)^2 = 4nh^2 d^2 + 2n^2 hd \tag{16}$

The feed-forward layer typically consists of two MLP layers. The first layer projects the hidden dimension $hd$ to the intermediate dimension $4hd$, while the second layer maps the intermediate dimension back to the hidden dimension $hd$. It involves the following computation:

$$\textbf{First layer: } n \times hd \times 4hd = 4n(hd)^2$$

$$\textbf{Second layer: } n \times 4hd \times hd = 4n(hd)^2$$

$$\textbf{Full feed foward layer: } 4n(hd)^2 + 4n(hd)^2 = 8nh^2 d^2 \tag{17}$$

As a consequence, the calculation of a complete transformer layer is summarized as follow:

$$\textbf{Full transformer layer: } 4nh^2d^2 + 2n^2hd + 8nh^2d^2 = 12nh^2d^2 + 2n^2hd \qquad (18)$$

The computation of each transformer layer is primarily influenced by the feed-forward layer, when the sequence length $n < 2hd$:

$$\textbf{Dominated by linear layer: } 4nh^2d^2 + 2n^2hd < 8nh^2d^2 \iff n < 2hd$$

While the computation of self-attention mechanism surpasses that of the feed-forward layer ($n > 2nd$), it is not reliable to claim that the quadratic term significantly hampers execution efficiency. This is because the computation of self-attention involves both a linear term and a quadratic term. The computation of the transformer layer is primarily influenced by the quadratic term when $n > 6hd$.

$$\textbf{Dominated by quadratic term: } 2n^2hd > 12nh^2d^2 \iff n > 6hd \qquad (19)$$

Consequently, the time complexity of the transformer layer is determined by the sequence length and hidden dimension:

$$\textbf{Dominated by } \begin{cases} \text{Feed-forward layer } 8nh^2d^2, & \text{if } n \le 2nd \\ \text{Overall linear term } 12nh^2d^2, & \text{if } 2nd < n \le 6nd \\ \text{Quadratic term } 2n^2hd, & \text{if } n > 6nd \end{cases} \qquad (20)$$

Considering the commonly used hidden dimension of 512, the computation is dominated by quadratic complexity when sequences exceed a length of 3072.

## B.4 Equivalence of Three Forward-pass Retention

According to Section 3.2, the parallel forward-pass is equivalent to the recurrent forward-pass. With the initial state variable $\boldsymbol{S}_0 = 0$, the recurrent forward-pass can be expressed as follows:

$$\textbf{Recurrent Form: } \boldsymbol{S}_n = \underbrace{\boldsymbol{K}_n^\top \boldsymbol{V}_n}_{\text{Single-token}} + \gamma \boldsymbol{S}_{n-1}, \quad \text{Retention}(\boldsymbol{X}_n) = \boldsymbol{Q}_n \boldsymbol{S}_n$$

$$\implies \boldsymbol{S}_n = \sum_{i=1}^{n} \gamma^{n-i} \boldsymbol{K}_i^\top \boldsymbol{V}_i, \quad \text{Retention}(\boldsymbol{X}_n) = \boldsymbol{Q}_n \sum_{m=1}^{n} \gamma^{n-m} \boldsymbol{K}_m^\top \boldsymbol{V}_m \qquad (21)$$

where the Retention($\boldsymbol{X}_n$) calculates the Retention at single-time $n$ by considering timestep $i$ up to the current time. It corresponds to the $n$-th timestep (row) of parallel forward-pass of Retention.

$$\textbf{Recurrent Form: } \text{Retention}(\boldsymbol{X}_n) = \boldsymbol{Q}_n \sum_{m=1}^{n} \gamma^{n-m} \boldsymbol{K}_m^\top \boldsymbol{V}_m$$

$$\implies \textbf{Parallel Form: } \text{Retention}(\boldsymbol{X}_n) = \underbrace{\boldsymbol{Q}_n}_{1 \times d_{qk}} \underbrace{\boldsymbol{K}_{m \le n}^\top}_{d_{qk} \times n} \odot \underbrace{\boldsymbol{D}_{m \le n}}_{n \times n} \underbrace{\boldsymbol{V}_{m \le n}}_{n \times d_v} \qquad (22)$$

When the recurrent forward-pass traverses all timesteps, the parallel and recurrent forward-passes of Retention become identical.

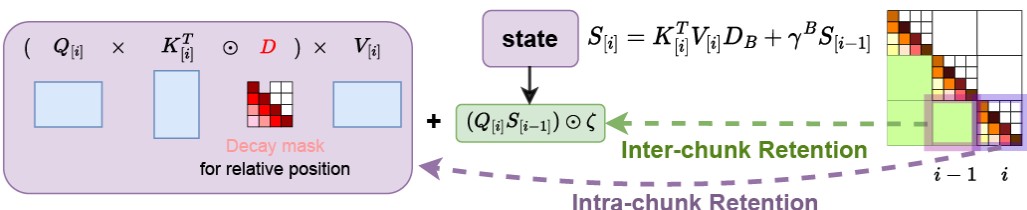

Figure 6: The chunk-wise forward-pass of Retention mechanism. It can be represented as parallel inner-chunk information (highlighted as purple) as well as recurrent inter-chunk information (highlighted as green) with the help of state variable $\boldsymbol{S}$.

Given the parallel and recurrent forward-passes of Retention, we aim to demonstrate the equivalence between chunk-wise forward-pass and the parallel and recurrent forward-passes. The chunk-wise Retention involves parallel intra-chunk and recurrent inter-chunk computation.

**Chunk-wise Form:** $\text{Retention}(\boldsymbol{X}_{[i]}) = \underbrace{(\boldsymbol{Q}_{[i]}\boldsymbol{K}_{[i]}^{\top} \odot \boldsymbol{D})\boldsymbol{V}_{[i]}}_{\text{Intra-chunk}} + \underbrace{(\boldsymbol{Q}_{[i]}\boldsymbol{S}_{[i-1]}) \odot \zeta}_{\text{Inter-chunk}}, \quad \zeta_j = \gamma^j$

$$\boldsymbol{S}_{[i]} = \underbrace{\boldsymbol{K}_{[i]}^{\top}\boldsymbol{V}_{[i]} \odot \boldsymbol{D}_B}_{\text{Current chunk}} + \underbrace{\gamma^B \boldsymbol{S}_{[i-1]}}_{\text{Past chunk}} \tag{23}$$

where $\zeta = [\gamma^1, \gamma^2, \ldots, \gamma^B]^{\top}$ is a column-vector of time-decay scaling factor for inter-chunk attention between the current chunk $[i]$ and the previous chunk $[i-1]$. Specifically, $\gamma^j$ is the scaling factor for the $j^{th}$ row of chunk $[i]$ from the last row of chunk $[i-1]$ such that the bigger the $j$ index the smaller the $\gamma^j$ value. Therefore, Retention recursively aggregates information from the $i$-th chunk (i.e., intra-chunk embedding) and the previous chunk (i.e., inter-chunk embedding).

For the per-chunk state variable $\boldsymbol{S}_{[i]}$, it computes current-chunk information as well as past-chunk information. The current-chunk information $\boldsymbol{K}_{[i]}^{\top}\boldsymbol{V}_{[i]}$ decays by $\boldsymbol{D}_B$, which is the last row of decay matrix $\boldsymbol{D}$. The past chunk information $\boldsymbol{S}_{[i-1]}$ is decayed with respect to the chunk size $B$. The initial state variable $\boldsymbol{S}_{[i=0]} = 0$ is computed recurrently given the chunk size $B$:

$$\boldsymbol{S}_{[i]} = \boldsymbol{K}_{[i]}^{\top}\boldsymbol{V}_{[i]} \odot \boldsymbol{D}_{B-1} + \gamma^B \boldsymbol{S}_{[i-1]} = \sum_{m=1}^{B} \gamma^{B-m}\boldsymbol{K}_m^{\top}\boldsymbol{V}_m + \gamma^B \boldsymbol{S}_{[i-1]} \tag{24}$$

Moreover, the update of state variable $\boldsymbol{S}_{[i]}$ can be reformulated in parallel form. The first term represents the information of current chunk, and the second term represented the past-chunk information decayed by the chunk size $B$. Consequently, $\boldsymbol{S}_{[i-1]}$ represents the state information from the beginning to the $(i-1)$-th chunk, and we represent the inter-chunk information in chunk-wise Retention as follows:

$$\boldsymbol{S}_{[i-1]} = \sum_{m=1}^{B*i} \gamma^{B*i-m}\boldsymbol{K}_m^{\top}\boldsymbol{V}_m = \boldsymbol{K}_{1:(B*i)}^{\top} \odot \boldsymbol{D}_{1:(B*i)}\boldsymbol{V}_{1:(B*i)}$$

$$\underbrace{(\boldsymbol{Q}_{[i]}\boldsymbol{S}_{[i-1]}) \odot \zeta}_{\text{Inter-chunk}} = (\boldsymbol{Q}_{(B*i):(B*(i+1))}\boldsymbol{K}_{1:(B*i)}^{\top} \odot \boldsymbol{D}_{1:(B*i)}\boldsymbol{V}_{1:(B*i)}) \odot \zeta$$

$$= \boldsymbol{Q}_{(B*i):(B*(i+1))}\boldsymbol{K}_{1:B*i}^{\top} \odot \boldsymbol{D}_{(B*i):(B*(i+1))}\boldsymbol{V}_{1:(B*i)} \tag{25}$$

where the intra-chunk computation updates each row of the lower triangular matrix (highlighted as green in Fig. 6). Together, the recurrent intra-chunk computation with the parallel intra-chunk computation (highlighted as purple 6) completes the chunk-wise forward-pass of Retention.

### B.5 TIMELYGPT PRE-TRAINING OVERFLOW

In the TimelyGPT pre-training, we illustrate the full processes of input processing, model training, and next-token prediction in Fig. 7. For a time-series input with $T$ timesteps and $V$ variates, it is tokenized via a convolution-subsampling module. This tokenizer, typically comprising two 1-D convolution layers with a kernel size of 3 and stride of 2. It produces tokens of shape $L \times V$, effectively reducing the sequence length to 1/4. These tokens are then transformed into input embeddings of shape $L \times d$ using an input projection, where $d$ is hidden dimension. The model training process involves a series of decoder layers, which takes segmented mulitple-chunk input embedding as input. Finally, the output embeddings with shape $L \times d$ are passed through an output projection layer to yield tokens of dimension $L \times V$ for the next-token prediction task.

## C ADDITIONAL INFORMATION FOR EXPERIMENTS

### C.1 DATASET DESCRIPTION

**Sleep-EDF dataset for Forecasting.** Sleep-EDF dataset proposed by (Kemp et al., 2000) from PhysioBank (Goldberger et al., 2000) contains sleep cassette data obtained from 153 patients. The

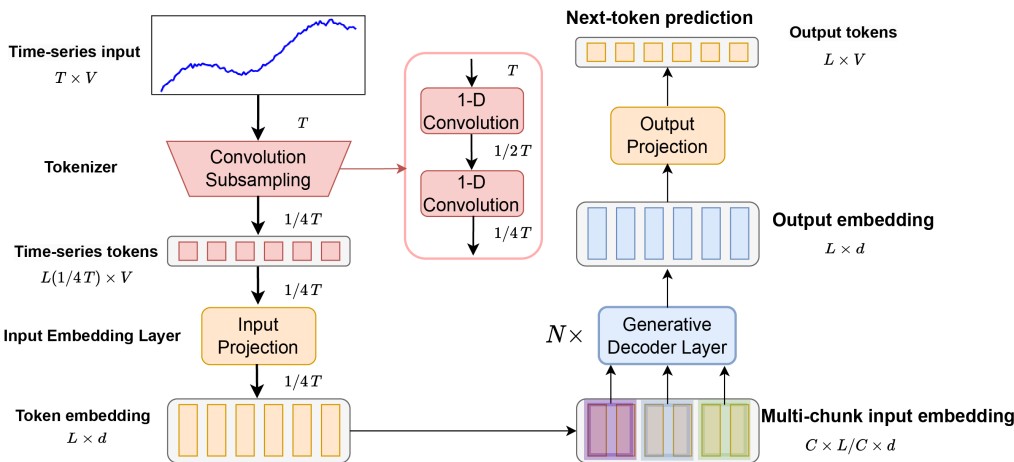

Figure 7: Schematic of the TimelyGPT Pre-Training Process

collected whole-night polysmnographic (PSG) sleep recordings encompass 7 variates: Electroen-cephalogram (EEG) (from Fpz-Cz and Pz-Oz electrode locations), electrooculogram (EOG) (horizontal), submental chin electromyogram (EMG), and an event marker. EEG and EOG signals were sampled at 100 Hz, while the EMG and event marker were sampled at 1 Hz. The sleep patterns correspond to the PSGs consist of five sleep stages: Wake (W), Non-rapid eye movement (N1, N2, N3) and Rapid Eye Movement (REM). In this problem, we aim to forecast signals across all 7 variates in the dataset.

**Sleep-EDF (sleep stage) dataset for Classification.** We leveraged the Sleep-EDF dataset for the classification task. In line with conventional practices, we selected the single EEG channel that captures signals from the Fpz-Cz electrode location. Since the Sleep-EDF dataset labels five sleep stages (i.e., W, REM, N1, N2 and N3), we only focused on EEG signals associated with these sleep stages, resulting in a total of 586.4 million timesteps.

**Epilepsy Seizure Classification.** Epileptic Seizure Recognition dataset (Andrzejak et al., 2001) comprises EEG measurements from 500 subjects. This dataset captures brain electrical activity from different regions and states, and are divided into segments of 23.6 seconds. The original dataset consists of five classes (eyes open, eyes closed, EEG measured in healthy brain region, EEG measured where the tumor was located and EEG measured on subject experiencing seizure episode). The first four classes are merged into a single class as these classes are unrelated to epileptic seizure, enabling a binary classification of epileptic seizures.

**PTB-XL Classification.** Physikalisch Technische Bundesanstalt large scale cardiology database (PTB-XL) (Alday et al., 2020) from PhysioBank (Goldberger et al., 2000) contains 21837 clinical 12-lead ECG signals (male: 11,379 and female: 10,458) with a duration of 10 seconds each, sampled with a rate of 500 Hz. These signals are categorized based on a set of twelve leads (I, II, III, AVL, AVR, AVF, V1, V2, V3, V4, V5, V6) with reference electrodes on the right arm are provided for each signal, resulting in twelve labels for classification.

**EMG Classification.** EMG dataset from PhysioBank (Goldberger et al., 2000) captures the electrical activity resulting from neural stimulation of muscles. This dataset provides insights about muscle functionality and the corresponding nerves responsible for their control. This dataset contains single-channel EMG signals sampled with a rate of 4 KHz. EMG recordings were obtained from the tibialis anterior muscle of volunteers exhibiting different degrees of muscular and neural disorders, resulting in three classification labels.

**PPG-Dalia Regression.** PPG-Dalia dataset (Reiss et al., 2019a) is available from Monash University, UEA&UCR Time Series Regression Archive (Tan et al., 2021). This dataset captures photoplethysmograph (PPG) data for motion compensation and heart rate estimation. Data was collected from 15 subjects engaging in daily life activities using both wrist-worn (Empatica E4) and chest-worn (RespiBAN Professional) devices. All signals were sampled at 700 Hz. The ECG recording

Table 6: Description of datasets utilized for forecasting, classification, and regression tasks.

| Details | Task | Variates | Timesteps | Samples | length | Classes |
|---|---|---|---|---|---|---|
| Sleep-EDF (Kemp et al., 2000) | Forecasting | 7 | 1.2B | None | None | None |
| Sleep-EDF (stage) (Kemp et al., 2000) | Classification | 1 | 586.4M | 195.5K | 3000 | 5 |
| Epilepsy | Classification | 1 | 2.0M | 11,500 | 178 | 2 |
| PTB-XL | Classification | 12 | 109.2M | 21,837 | 5000 | 5 |
| EMG | Classification | 1 | 306.0K | 204 | 1500 | 3 |
| PPG-Dalia | Regression | 4 | 16.6M | 64697 | 256 & 512* | None |
| IEEEPPG | Regression | 5 | 3.1M | 3096 | 1000 | None |
| RR | Regression | 2 | 31.5M | 7870 | 4000 | None |
| HR | Regression | 2 | 31.8M | 7949 | 4000 | None |
| SpO2 | Regression | 2 | 31.8M | 7949 | 4000 | None |

* The sequence length of the PPG-Dalia dataset varies across different dimensions.

serve as a ground truth for heart rate. The PPG and 3D-accelerometer data are used to estimate heart rate, while accounting for motion artefacts.

**IEEEPPG Regression.** IEEEPPG dataset (Zhang et al., 2015) is sourced from Monash University, UEA&UCR Time Series Regression Archive (Tan et al., 2021). This dataset aims to estimate heart rate by utilizing PPG and ECG signals. The dataset includes two-channel PPG signals, three-axis acceleration signals, and one-channel ECG signals. These signals were recorded simultaneously and sampled at a rate of 125 Hz.

**RR, HR and SpO2 Regression.** BIDMC Respiratory Rate (RR), heart rate (HR) and blood oxygen saturation level (SpO2) datasets (Pimentel et al., 2017) are available at Monash University, UEA&UCR Time Series Regression Archive (Tan et al., 2021). These datasets were obtained from the Physionet's BIDMC PPG and Respiration dataset (Goldberger et al., 2000). They contain PPG and ECG signals sampled at a rate of 125 Hz, which are designed for estimating RR, HR, and SpO2.

**PopHR database.** The database hosts a massive amount of longitudinal heterogeneous claim data from the provincial government health insurer in Quebec, Canada (Régie de l'assurance maladie du Québec, RAMQ) on health service use Shaban-Nejad et al. (2016); Yuan et al. (2018). In total, there are approximately 1.3 million participants in the PopHR database, which represents a randomly sampled 25% of the population in the metropolitan area of Montreal between 1998 and 2014. Cohort membership is maintained dynamically by removing deceased residents and actively enrolling newborns and immigrants.

## C.2 DATA PRE-PROCESSING

For forecasting experiment, we utilized the entire seven-variates Sleep-EDF dataset, excluding the instances with unkown sleep stages. We divided the dataset into training (80%), validation (10%) and testing (10%) sets. We used the whole train set for pre-training and 20% of train set for fine-tuning. During pre-training, TimelyGPT is pre-trained on non-overlapped sequences with 4,000 timesteps. To use Sleep-EDF dataset for classification, we only utilized the single EEG (Fpz-Cz electrode signal) channel and considered sleep stage from the Sleep-EDF dataset. All datasets utilized for classification and regression tasks are are divided into training (85%), validation (5%), and testing (10%) sets.

For the irregularly-sampled time series on the PopHR database, we converted ICD-9 codes to Phe-Codes using the expert-defined PheWAS catalog Denny et al. (2013; 2010). We calculated the prevalence ratio for each PheCode in the true patient population and the general population for 12 disease labels. For each disease label, we chose PheCodes that either had a prevalence ratio exceeding 5 or were among the top 5 most prevalent. Furthermore, we excluded patients with fewer than 10 occurrences, resulting in a processed dataset of 47,000 patients and 2.2 million records. The processed dataset is then divided into training (80%), validation (10%), and testing (10%) sets. In total, 133 unique PheCodes were included in our experiment. Given that diagnoses are discrete value,

there was no need to utilize the convolution-subsampling tokenizer. Furthermore, we specified a patch size of 2 for PatchTST, which differentiates it from the TST model.

### C.3    EXPERIMENT DESIGN FOR TRANSFORMER'S SCALING LAW IN TIME-SERIES

To evaluate the scaling law of Transformers in the context of time-series, we trained TimelyGPT, Informer, and DLinear from scratch using the SleepEDF dataset. Both the look-up and forecasting windows were configured to 256 timesteps. We selected specific timesteps ranging from 10e5 to 10e9, dividing into training (80%), validation (10%), and testing (10%) sets.

**DLinear model.**    This model is configured with a 256-timestep look-up and forecasting window, maintaining channel independence.

**TimelyGPT.**    This architecture comprises 10 decoder layers. The hidden dimensions for queries, keys, values, and feed-forward layers are configured at 192, 192, 384, and 768 respectively.

**Informer.**    This model is structured with 8 encoder layers and 2 decoder layers. The hidden dimensions for model and feed-forward layer are set as 224 and 896.

### C.4    PRE-TRAINING AND FINE-TUNING STRATEGIES FOR TIMELYGPT AND BASELINES

TimelyGPT employs a per-training paradigm for unlabeled time-series data base on next-token prediction task. Given that each sequence with a [SOS] token, this setup enables model to predict subsequent tokens by shifting the sequence rightwards (Chen et al., 2020). In the final layer, each token's output representation is fed into a linear layer for next-token prediction using cross-entropy loss. Following PatchTST's findings that end-to-end fine-tuning yields optimal results (Nie et al., 2023), we adopt end-to-end fine-tuning on the pre-trained TimelyGPT. The average of hidden representation in the final layer is fed into the linear layer for downstream tasks. The model undergoes self-training for 20 epochs on datasets outlined in Section. 4.2, Section. 4.3, and Section. 4.4, incorporating early stopping. Once the PTM on each dataset is available, we employ end-to-end fine-tuning for 5 epochs.

For encoder-only time-series transformer, PatchTST adopts a pre-training technique with masked self-supervised learning and non-overlapped patches (Nie et al., 2023). Moreover, PatchTST considers high masking ratio where 40% of the patches are masked with zero values. In contrast, encoder-decoder transformer baselines such as Informer, Autoformer, Fedformer lack pre-training representation learning. We implement the masking strategy used in (Zerveas et al., 2020; Nie et al., 2023), where 40% of timesteps are randomly masked. For these transformer baselines, we apply same pre-training and fine-tuning procedures as TimelyGPT.

### C.5    EXPERIMENT DETAILS FOR TIMELYGPT AND BASELINES

Given the demonstrated scaling law in Section 4.1, we tailored the hyperparameters and model parameters of TimelyGPT and all transformer baselines according to the dataset sizes: 18 million model parameters for forecasting and classification tasks on SleepEDF dataset; 7.5 million model parameters for classification and regression tasks using PTB-XL and PPGDalia datasets; 3 million model parameters for classification task of irregularly-sampled time-series. The details of model parameters and architectures of TimelyGPT and all transformer baslines are shown in Table 7 and Table 8, respectively.

Table 7: Configurations of TimelyGPT model on different experiments for diverse dataset

| Datasets | Sleep-EDF | PTB-XL & PPGDalia | PopHR |
|---|---|---|---|
| Experiments | Forecasting classification | classification Regression | classification |
| Data Size (timesteps) | 1.2B | 109.2M & 16.6M | 8.4M |
| Fine-tuning task type | Seq2seq & Seq2vec | Seq2vec & Seq2vec | Seq2vec |
| Model Parameters | 18M | 7.5M | 3M |
| Decoder Layers | 12 | 8 | 8 |
| Heads | 8 | 8 | 4 |
| Dim (Queries) | 320 | 240 | 144 |
| Dim (Keys) | 320 | 240 | 144 |
| Dim (Values) | 640 | 480 | 288 |
| Dim (FF Layers) | 640 | 480 | 288 |

Table 8: Configurations of all transformer baselines on different experiments for diverse datasets

| Datasets | Sleep-EDF | PTB-XL & PPGDalia | PopHR |
|---|---|---|---|
| Experiments | Forecasting classification | classification Regression | classification |
| Data Size (timesteps) | 1.2B | 109.2M & 16.6M | 8.4M |
| Fine-tuning task type | Seq2seq & Seq2vec | Seq2vec & Seq2vec | Seq2vec |
| Model Parameters | 18M | 7.5M | 3M |
| Enc-Dec (Enc) Layers | 6-6 (12) | 4-4 (8) | 4-4 (8) |
| Heads | 8 | 8 | 4 |
| Dim (Queries) | 384 | 288 | 144 |
| Dim (Keys) | 384 | 288 | 144 |
| Dim (Values) | 384 | 288 | 144 |
| Dim (FF Layers) | 1536 | 1152 | 576 |

## C.6 Ultra-long-term Forecasting Results on Sleep-EDF Dataset

Table 9: Comparison of TimelyGPT as well as 7 baselines for ultra-long-term forecasting experiment on large-scale SleepEDF dataset.

| Window Size | 720 | 2000 | 6000 |
|---|---|---|---|
| TimelyGPT | 0.542 | **0.567** | **0.575** |
| Informer | 0.675 | 1.013 | 1.256 |
| Autoformer | 0.532 | 0.908 | 1.026 |
| Fedformer | 0.515 | 0.865 | 0.912 |
| PatchTST | **0.456** | 0.768 | 0.824 |
| DLinear | 0.521 | 0.840 | 0.929 |
| TS2Vec | 0.602 | 1.231 | 1.204 |
| TimesNet | 0.471 | 0.742 | 0.865 |
| GPT-2 + RoPE | 0.516 | 0.583 | 0.715 |
| GPT-2 | 0.525 | 0.815 | 1.072 |

## C.7 Classification Results of Irregularly-sampled Time Series Experiment

The PopHR database includes 12 predefined disease labels: Acute Myocardial Infarction (AMI), Asthma, Congestive Heart Failure (CHF), Chronic Obstructive Pulmonary Disease (COPD), Dia-

Table 10: Comparison of TimelyGPT as well as 9 baselines for multi-label classification (AUPRC % on 12 phenotype labels.

| | AMI | Asthma | CHF | COPD | Diabetes | HTN | Schizo | IHD | HIV | Epilepsy | Autism | ADHD |
|---|---|---|---|---|---|---|---|---|---|---|---|---|
| TimelyGPT | 0.612 | 0.568 | 0.637 | 0.541 | 0.563 | 0.586 | 0.551 | 0.584 | 0.549 | 0.560 | 0.576 | 0.583 |
| PatchTST | 0.561 | 0.516 | 0.585 | 0.489 | 0.512 | 0.535 | 0.500 | 0.533 | 0.497 | 0.509 | 0.524 | 0.531 |
| TST | 0.569 | 0.524 | 0.593 | 0.497 | 0.520 | 0.543 | 0.508 | 0.541 | 0.505 | 0.517 | 0.532 | 0.539 |
| CRT | 0.516 | 0.472 | 0.541 | 0.445 | 0.467 | 0.490 | 0.455 | 0.488 | 0.453 | 0.464 | 0.480 | 0.487 |
| TS-TCC | 0.502 | 0.457 | 0.526 | 0.430 | 0.453 | 0.476 | 0.441 | 0.474 | 0.438 | 0.450 | 0.465 | 0.472 |
| TS-T | 0.480 | 0.436 | 0.505 | 0.409 | 0.431 | 0.454 | 0.419 | 0.452 | 0.417 | 0.428 | 0.444 | 0.451 |
| mTAND | 0.607 | 0.557 | 0.616 | 0.547 | 0.553 | 0.571 | 0.561 | 0.574 | 0.558 | 0.539 | 0.544 | 0.542 |
| GRU-D | 0.584 | 0.560 | 0.599 | 0.533 | 0.495 | 0.599 | 0.513 | 0.586 | 0.547 | 0.552 | 0.558 | 0.565 |
| SeFT | 0.524 | 0.479 | 0.548 | 0.452 | 0.475 | 0.498 | 0.463 | 0.496 | 0.460 | 0.472 | 0.487 | 0.494 |
| RAINDROP | 0.557 | 0.579 | 0.582 | 0.516 | 0.528 | 0.531 | 0.526 | 0.549 | 0.524 | 0.525 | 0.560 | 0.578 |

betes, Hypertension (HTN), Schizophrenic disorders (Schizo), Ischemic Heart Disease (IHD), Human immunodeficiency virus (HIV), Epilepsy, Autism, Adult attention-deficit/hyperactivity disorder (ADHD). The full results (AUPRC) of multi-label classification on PopHR's irregularly-sampled time series are detailed in Table 10

## C.8   RESULTS OF ABLATION STUDIES

Table 11 shows the complete results of ablation studies on the downstream classification tasks using the PTM models. We pre-trained on continuous bio-signals from Sleep-EDF and irregularly-sampled time-series from PopHR database (CHF phenotype only), and then fine-tuned for classification tasks. In the ablation studies, we selectively **omitted** the specific components in the proposed TimelyGPT. Additionally, removing all the components results in the GPT-2 baseline. It should be noted that the diagnoses in PopHR's irregularly-sampled time series is discrete and do not necessitate the use of tokenizer (i.e., the Convolution Subsampling module).

Table 11: Results of ablation studies using downstream classification tasks on the Sleep-EDF and PopHR-CHF data. We remove specific components by indicating w/o.

| Method / Classification | Sleep-EDF | PopHR-CHF |
|---|---|---|
| TimelyGPT (with Pre-training) | 89.21 | 64.89 |
| w/o Convolution Subsampling | 88.65 | — |
| w/o Temporal Convolution | 87.65 | 63.21 |
| w/o Exponential Decay | 86.52 | 55.36 |
| w/o Relative RoPE (GPT-2) | 84.25 | 51.83 |
| **TimelyGPT (w/o Pre-training)** | **85.32** | **64.06** |

Table 12: Comparison of different configurations of convolutional operators in Temporal Convolution module

| Configuration | Accuracy (%) |
|---|---|
| Depth-wise Conv + Point-wise Conv (**Ours**) | 89.21 |
| Point-wise Conv + Depth-wise Conv + Point-wise Conv (Gulati et al. (2020)) | 88.75 |
| Depth-wise Conv (Wu et al. (2020)) | 88.12 |
| Point-wise Conv | 87.56 |
| w/o Temporal Convolution | 87.03 |

## C.9   HYPERPARAMETER ANALYSIS OF CONVOLUTION MODULE

In TimelyGPT, we propose a depth-wise separable convolution for the temporal convolution module, consisting both depth-wise and point-wise convolution. For comparison, we evaluated two

alternative configurations: (1) depth-wise convolution only Wu et al. (2020); (2) a combination of point-wise convolution, depth-wise convolution, and another point-wise convolution Gulati et al. (2020). Additionally, we assessed assessed the impact of using only point-wise convolution. These configurations were compared based on classification accuracy in downstream classification task using the SleepEDF dataset Table 12.

### C.10 VISUALIZATION FOR EXTRAPOLATION

We visualized ultra-long-term forecasting for TimelyGPT alongside the best baseline PatchTST, DLinear, and ablated methods (vanilla GPT-2 and GPT-2 with RoPE embedding). Focusing on an example of sleep stage transition, we used a 2000-timestep look-up window as a prompt and a 6000-timestep forecasting window for prediction. In these visualizations, forecasting results beyond 4000 timesteps are considered as extrapolation since they exceed the 4000-timestep training length.

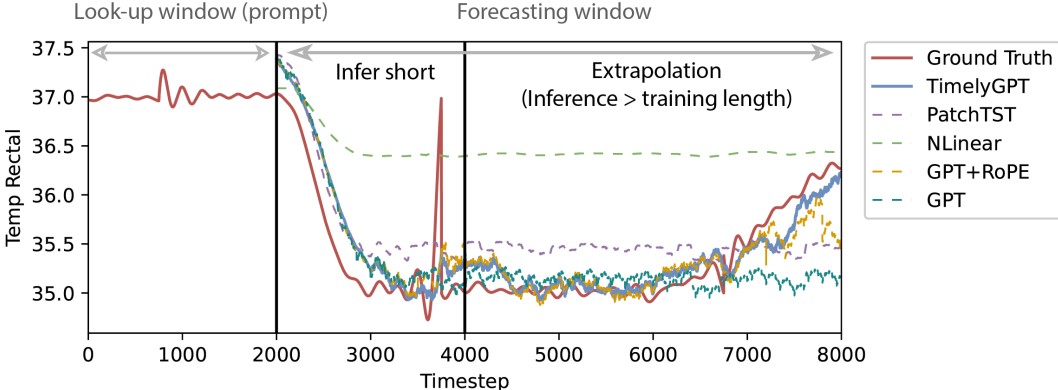

Figure 8: Example of forecasting experiments on the trend signal (rectal temperature)

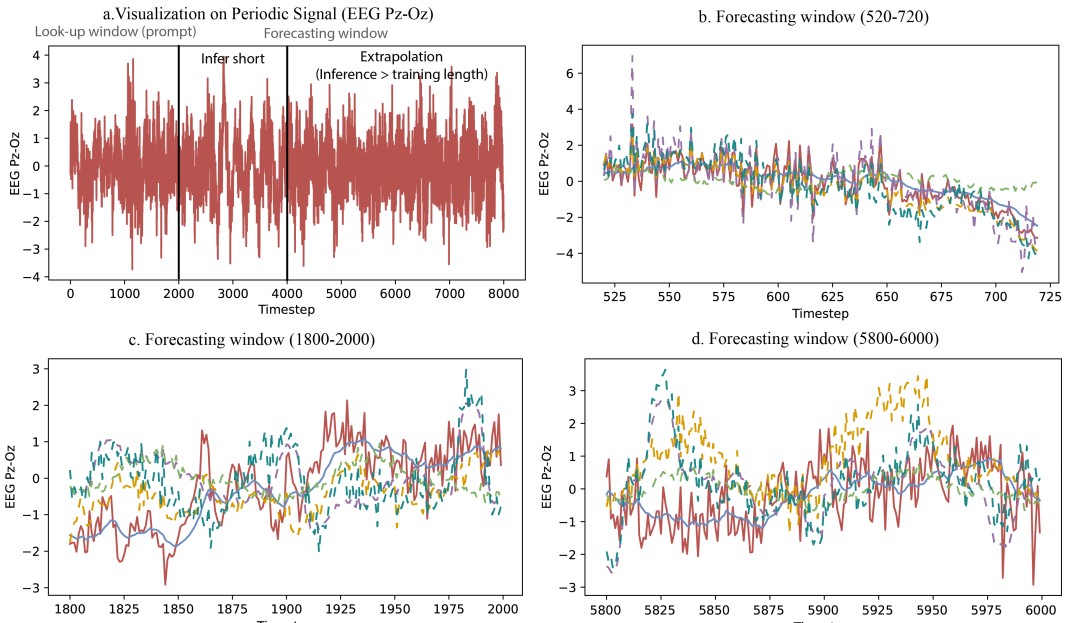

Figure 9: Example of forecasting experiments on the period signal (EEG Pz-Oz). Panel a indicates the groundtruth of EEG Pz-Oz singal, where the remaining panels indicate the different forecasting windows. The x-axis was offset by 2000 observation window.

In the forecasting of trend biosignal (Fig. 10), TimelyGPT with xPOS Embedding demonstrates superior extrapolation accuracy. Its forecast aligns well with the groundtruth for the 6000 timesteps

and captures distinct trend pattern. TimelyGPT's extrapolation ability surpasses GPT-2 with RoPE embedding, which primarily excels in processing periodic patterns. Similar to PatchTST, hindered by its dependency on linear mapping (Li et al., 2023), vanilla GPT-2 exhibits limited forecasting performance beyond 1000 timesteps. In the forecasting of periodic biosignal (Fig. 9), TimelyGPT generate smooth forecast, which aligns well with the groundtruth. In contrast, GPT-2 with RoPE embedding yields oscillatory outputs, which emphasizes periodic patterns and compromises its extrapolation potential.

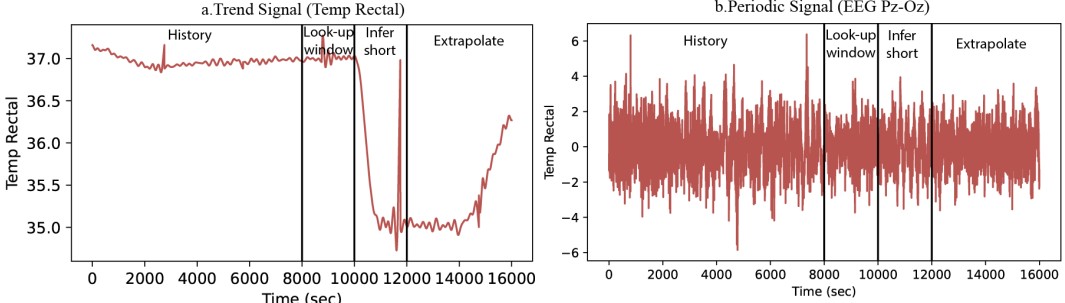

Figure 10: Example of forecasting experiments on the trend signal (rectal temperature) and the period signal (EEG Pz-Oz) with 8,000-timestep history.

