# OpenReview forum: "TimelyGPT: Recurrent Convolutional Transformer for Long Time-series Representation"
_ICLR.cc/2024/Conference — Submitted to ICLR 2024_

### Official Review · Reviewer_JDYd · 2023-11-01

**Soundness:** 3 good
**Presentation:** 2 fair
**Contribution:** 3 good
**Rating:** 5
**Confidence:** 4

**Summary:**

This paper proposes TimelyGPT, a new transformer architecture for long-time series representation and modeling. For the embedding, xPOS embedding segments long sequences and uses exponential decay based on relative distances to enable extrapolation. For the representation, recurrent attention retention mechanism models global context by integrating short and long-term memory; convolution modules applied on inputs (tokenizer) and hidden states capture local interactions; and ecoder-only autoregressive design allows generating representations for future unseen timesteps. The authors demonstrate the effectiveness of TimelyGPT in modeling both continuously monitored biosignal data and irregularly-sampled time-series data, which are common in longitudinal electronic health records. They advocate for a shift in time-series deep learning research from small-scale modeling from scratch to large-scale pre-training.

**Strengths:**

Originality:

The integration of relative position modeling, recurrent attention, and convolutions for transformers is novel and creative.

Applying these architectures specifically for long-time series modeling is an original contribution.

The idea of decoding future representations and extrapolation is innovative.

Significance:

The work moves transformer research for time series modeling in an important new direction.

The extrapolation capabilities and integrated components can positively impact many applications.

The ability to handle long irregular sequences has high relevance for domains like healthcare.

Clarity:

The paper is clearly structured with the techniques, experiments, results, and analyses presented in a logical flow.

The writing clearly explains the intuition and technical details of the methods.

Tables and figures help summarize key quantitative results.

**Weaknesses:**

Analysis of the computational overhead and scaling behavior compared to other transformers could be expanded.

Only one large-scale time series dataset (Sleep-EDF) was used for pre-training experiments. More diverse pre-training data could help.

For the classification tasks, comparison to a broader range of datasets and architectures like RNNs would be useful. For the forecasting, what's the performance for the ETT, weather and electricity dataset used in the PatchTST paper.

Theoretical analysis or frameworks to relate the different components to time series properties is limited.

Hyperparameter sensitivity analysis could provide more insights into optimal configurations.

Evaluating true few-shot generalization with limited fine-tuning data could better demonstrate transfer learning benefits.

The focus is on a decoder-only architecture. Encoder-decoder and encoder-only architectures could also be analyzed.

Societal impacts regarding potential misuse of forecasting need to be considered.

**Questions:**

While this paper explores an intriguing aspect of time-series analysis—specifically, modeling irregular time-series—it is not without its shortcomings, particularly in the design of experiments meant to objectively situate this work within its field. Please refer to the 'weaknesses' section for a detailed list of concerns raised by the reviewer. The reviewer would be pleased to revise their score upwards provided these issues are adequately addressed.

---

> ### Author Response · Authors · 2023-11-21
> **Response to Reviewer JDYd (Q1, Q2, Q3, Q4)**
>
> Thanks for your careful review and constructive suggestions. Please find our response to your questions and concerns.
>
> **Q1: Analysis of the computational overhead and scaling behavior compared to other transformers could be expanded.**
>
> R1: Our study primarily examines computational complexities, emphasizing TimelyGPT's linear training complexity for long sequence length. As established in Section 2.5, the quadratic term becomes dominant when $n > 6hd$ (where $n$ is sequence length, $h$ is head number, $d$ is embedding size per head).
> Leveraging chunk-wise retention mechanism, TimelyGPT segments each long sequence into multiple small chunks, where $n < 6hd$ in each chunk. Therefore the time complexity is mainly on the embedding size, i.e., the second term in $O(12n h^2 d^2 + 2n^2hd)\equiv O(12n h^2 d^2)$, which is linear to the sequence length. Detailed analysis is provided in Section 2.5 and Appendix B.3.
>
> In practice, we analyzed the scaling law on the Sleep-EDF dataset (i.e., MAE as a function of dataset size; Fig. 2), which is the biggest time-series dataset publicly available. We plan to expand this analysis to larger datasets in further studies.
>
> **Q2: Only one large-scale time series dataset (Sleep-EDF) was used for pre-training experiments. More diverse pre-training data could help.**
>
> R2: Since Sleep-EDF is the biggest time-series dataset publicly available, we are limited in finding datasets of comparable size.
> Nonetheless, we also explore pre-training on other datasets like PTB-XL. This dataset has been examined in other pre-training studies (CRT and TF-C), offering a basis for comparative analysis for follow-up works.
>
> **Q3: For the classification tasks, comparison to a broader range of datasets and architectures like RNNs would be useful. For the forecasting, what's the performance for the ETT, weather and electricity dataset used in the PatchTST paper.**
>
> R3: To address the concern about a broader range of architectures, our revised manuscript now includes comparisons with LSTM (same layer and dim as TimelyGPT) and two novel approaches (TS2Vec and TimesNet) in classification and regression tasks. The table presented below shows TimelyGPT's superior performance, particularly over LSTM. This effectiveness can be attributed to TimelyGPT's recurrent attention mechanism, effectively combining the strengths of the sequential modeling capabilities of RNNs and the scalability of transformers.
>
> | Task | Classification (Accuracy %)  |  | |  | Regression (MAE)  |  |  |  |
> |------|--------------------------------------|----------|--------|-----|--------------------------|----|----|------|
> | Pre-training| Sleep-EDF | | PTB-XL|| PTB-XL & PPGDalia | | | |
> | Fine-tuning| Sleep-EDF | Epilepsy | PTB-XL | EMG | IEEEPPG | RR | HR | SpO2 |
> | **TimelyGPT** | **89.21**  | **92.79** | **86.52**  | **95.87** | **26.17**  | **2.78** | **8.53** | **4.26** |
> | TimesNet | 83.58 | 85.96 | 79.38 | 89.26 | 29.95 | 4.19 | 13.65 | 4.83 |
> | TS2Vec | 86.21 | 88.27 | 82.65 | 93.77 | 27.89 | 3.53 | 11.56 | 4.60 |
> | LSTM | 80.15 | 76.51 | 78.83 | 86.95 | 30.15 | 4.95 | 14.37 | 5.05 |
>
> In Section 2.1, we point out that commonly used forecasting datasets are small data (up to 69K). As demonstrated in Section 4.1, time-series transformers follow scaling law and underperform on small data. Consequently, we focus on large-scale pre-training (up to 1.2B) instead.
>
> **Q4: Theoretical analysis of frameworks to relate the different components to time series properties is limited.**
>
> R4: We consider the following time-series properties: (1) trend patterns; (2) sequential nature; (3) multi-scale signals (i.e., patterns revealed at sub-sequence of different lengths); (4) global and local features.
>
> Our TimelyGPT framework addresses the above four properties  with the three key components, namely xPOS, retention, and convolution.
> For xPOS, we examine its potential for time-series extrapolation, building on its known capability to model long-term dependencies in text data [1]. We found that xPOS captures long-term **trend patterns** in time-series, facilitating extrapolation over long timesteps.
> Additionally, RNNs are recognized for their effectiveness in sequential data modeling, aligning well with the inherent **sequential nature** of time-series data [2].
> Convolution modules are capable of extracting both **local features** [3] and **multi-scale features ** [4] from time-series data, representing various characteristics of time-series. The integration of attention mechanisms and convolution modules  effectively capture both **global and local interactions** in time-series data [5].
>
> [ref 1] A Length-Extrapolatable Transformer
>
> [ref 2] Efficiently modeling long sequences with structured state spaces
>
> [ref 3] Convolutional Networks for Images, Speech, and Time Series
>
> [ref 4] Omni-scale CNNs: a simple and effective kernel size configuration for time series classification
>
> [ref 5] Lite Transformer with Long-Short Range Attention

---

> ### Author Response · Authors · 2023-11-21
> **Response to Reviewer JDYd (Q5, Q6, Q7, Q8, Q9)**
>
> **Q5: Hyperparameter sensitivity analysis could provide more insights into optimal configurations.**
>
> R5: Our optimization of model parameters is guided by the scaling law discussed in Section 4.1. As illustrated in Fig. 2b, the largest TimelyGPT with 18M parameters continues to improve as data size grows. Therefore, our choice of the overall number of parameters such as number of layers and embedding size is mainly limited to our computational resource.
> For certain hyperparameters, like the decay rate, we follow the specifications from the RetNet paper.
> Additionally, we have conducted an analysis of convolution operators, the results of which are shown in Table 12.
>
> **Q6: Evaluating true few-shot generalization with limited fine-tuning data could better demonstrate transfer learning benefits.**
>
> R6: In the revised version, the Section 5 (Conclusion) discussed that the pre-trained TimelyGPT exhibits limited generalization ability across different types of biosignals. For example, model pre-trained on ECG (SleepEDF dataset) does not generalize well on PPG signal. We acknowledge this limitation in Section 5 and identify it as an avenue for future exploration.
>
> **Q7: The focus is on a decoder-only architecture. Encoder-decoder and encoder-only architectures could also be analyzed.**
>
> R7: Existing time-series transformer mainly focus on on encoder-only (like PatchTST) and encoder-decoder (like Informer, Autoformer, Fedformer) architectures, which have been investigated in experiments. Our paper focuses on decoder-only transformer since extrapolable embedding is constrained to unidirectional attention [1]. Attempts to adapt to bidirectional attention have shown reduced extrapolation performance [1]. Our future plans involve developing a novel design to accommodate extrapolable embeddings to bidirectional attention and encoder architecture.
>
> [ref 1] Train Short, Test Long: Attention with Linear Biases Enables Input Length Extrapolation
>
> **Q8: Societal impacts regarding potential misuse of forecasting need to be considered.**
>
> R8: We revise manuscript by incorporating societal impacts in Section. 5 (Conclusion). The misuse of pre-trained model in forecasting can lead to serious privacy risks, such as unauthorized prediction of patient health status and breaches of privacy.
>
> **Q9: While this paper explores an intriguing aspect of time-series analysis—specifically, modeling irregular time-series—it is not without its shortcomings, particularly in the design of experiments meant to objectively situate this work within its field. Please refer to the 'weaknesses' section for a detailed list of concerns raised by the reviewer. The reviewer would be pleased to revise their score upwards provided these issues are adequately addressed.**
>
> R9: In the revised version, we provide a discussion of key shortcomings in Section 5 (Conclusion). It covers domain adaptation, the constraint of xPOS to unidirectional attention, and social concerns regarding privacy leak.

---

> ### Author Response · Authors · 2023-11-23
> **Final Confirmation with Reviewer JDYd**
>
> Thank you for spending your time and effort to review our paper. We greatly appreciate your valuable suggestions and have made **revisions** accordingly.
>
> We understand that you initially had reservations about our work, and we would like to ensure that all your concerns have been adequately addressed. If you believe that your concerns have been resolved, we would be grateful if you could consider revising your rating. We highly value your feedback and sincerely appreciate your consideration in this matter.

---

> > ### Comment · Reviewer_JDYd · 2023-11-23
> > **Thanks for the response**
> >
> > Dear Authors,
> >
> > Thanks for the response. My major questions have been addressed. Please consider the remaining concerns raised by other reviewers to improve the presentation of this paper.
> >
> > Best.

---

> > > ### Author Response · Authors · 2023-11-23
> > > **Grateful Acknowledgment of Reviewer JDYd's Feedback**
> > >
> > > Thank you for acknowledging the revisions we made in response to your questions. We are dedicated to further improving our paper and presentation by addressing the concerns raised by other reviewers, as you've suggested.
> > >
> > > We noticed that another reviewer adjusted their rating post our revisions. In light of this, we are wondering if you might also consider reassessing your rating in view of our responses and improvements.Your reassessment would be incredibly valuable to us.

---

### Official Review · Reviewer_5JkM · 2023-11-01

**Soundness:** 3 good
**Presentation:** 3 good
**Contribution:** 2 fair
**Rating:** 6
**Confidence:** 5

**Summary:**

This paper first revisits time series transformers and identifies the shortcomings of previous studies. Based on the findings, this paper introduces a Timely Generative Pre-trained Transformer (TimelyGPT), which combines recurrent attention and temporal convolution modules to capture global and local temporal dependencies in long sequences. TimelyGPT is comprised of three major components: xPOS embeddings, retention module for irregularly sampled time series and a convolution model for local interaction.

**Strengths:**

1. This paper introduces an interesting TimelyGPT for long time series representation learning. A major advance is that TimelyGPT has a superior performance on ultra-long time series forecasting.

2. The proposed architecture seems technical sound. xPOS and retention module are capable of modeling long time series and the convolution module could effectively capture local information.

3. Experimental results on several larger-scale datasets show that the proposed TimelyGPT could outperform SOTA baselines.

**Weaknesses:**

1. Most of the empirical insights (section 2) for using Transformers for time series presented by this paper are kind of repeating the findings of previous works [1][2].

2. In the ablation studies, this paper only presents the results of classification and the results for forecasting tasks are not provided.

3. The model architecture is a little bit incremental. xPOS embeddings and retention module are simple extensions of previous works.

[1] Ailing Zeng, Muxi Chen, Lei Zhang, and Qiang Xu. Are transformers effective for time series forecasting?, 2022.

[2] Ofir Press, Noah A. Smith, and Mike Lewis. Train short, test long: Attention with linear biases enables input length extrapolation, 2022.

**Questions:**

1. How do you calculate the dataset size?

2. How does TimelyGPT perform if pre-training is removed?

3. For ultra-long-term forecasting (section 4.2),

3.a. How many parameters do baselines have?

3.b. Are the baselines pre-trained? How do you pre-train baselines?

---

> ### Author Response · Authors · 2023-11-21
> **Response to Reviewer 5JkM (W1, W2, W3, Q1, Q2, Q3)**
>
> We sincerely appreciate your thorough review and valuable feedback. We provide our responses to your insightful questions and concerns.
>
> **W1: Most of the empirical insights (section 2) for using Transformers for time series presented by this paper are kind of repeating the findings of previous works [1][2].**
>
> **[1] Ailing Zeng, Muxi Chen, Lei Zhang, and Qiang Xu. Are transformers effective for time series forecasting?, 2022.**
>
> **[2] Ofir Press, Noah A. Smith, and Mike Lewis. Train short, test long: Attention with linear biases enables input length extrapolation, 2022.**
>
> R1: Our research diverges from the conclusion of paper [1]. Rather than attributing transformer underperformance solely to model design [1], we identify overfitting as a critical issue in Section 2.1 and Section 4.1. Additionally, paper [1] believes that inherent flaws in attention and position embedding lead to underperformance. In our study, we investigate effective designs of attention and position embedding for time-series modeling in Section 2.2 and Section 2.3, respectively. Inspired by the findings of paper [2], we propose the extrapolation technique in time-series transformers for the first time.
>
> [ref 1] Are transformers effective for time series forecasting?
>
> [ref 2] Train Short, Test Long: Attention with Linear Biases Enables Input Length Extrapolation
>
> **W2: In the ablation studies, this paper only presents the results of classification and the results for forecasting tasks are not provided.**
>
> R2: In the revised version, we provide a forecasting-specific ablation studies in Table.9. The table reveals that vanilla GPT-2 lacks extrapolation ability; while GPT-2 with RoPE achieves limited extrapolation over shorter distances. Section 4.6 offers visualized ablation results, indicating xPOS effectively captures long-term trends and achieves superior extrapolation performance.
>
> | Window Size | 720 | 2000 | 6000 |
> |-------------|-----|------|------|
> | **TimelyGPT**   | 0.542 | **0.567** | **0.575** |
> | GPT2 + RoPE |  **0.516** | 0.583 | 0.715 |
> | GPT2        | 0.525 | 0.815 | 1.072 |
>
> **W3: The model architecture is a little bit incremental. xPOS embeddings and retention module are simple extensions of previous works.**
>
> R3: Although xPOS and extrapolation are applied in NLP domain, our paper is the first work to thoroughly investigate extrapolable embedding in time-series forecasting. Note that NLP and time-series domain have fundamental difference, such as trend and periodic patterns in time-series. We explore the underlying mechanism that enables xPOS's extrapolation in time-series domain, which is not studied in previous works. Additionally, TimelyGPT is the first work to introduce RNN-based transformer in time-series domain. We expect a research shifts toward unexplored areas like extrapolable embedding and RNN-based transformers in time-series domains.
>
> **Q1: How do you calculate the dataset size?**
>
> R1: The revised manuscript explains the dataset size in terms of timesteps clearly.
>
> **Q2: How does TimelyGPT perform if pre-training is removed?**
>
> R2: The revised version includes a quantitative ablation study on pre-training in Table. 11. The following table shows that TimelyGPT with pre-training obtains a notable increase of 3.89\% in biosignal classification (SleepEDF), while the improvement (0.83\%) is less evident in irregularly-sampled time series (PopHR).
>
> | Method / Classification | Sleep-EDF | PopHR |
> |-------------------------|-----------|-------|
> | **TimelyGPT (with Pre-training)** | 89.21 | 64.89 |
> | TimelyGPT (w/o Pre-training) | 85.32 | 64.06 |
>
> **Q3: For ultra-long-term forecasting (section 4.2),**
>
> **3.a. How many parameters do baselines have?**
>
> R3.a: For ultra-long-term forecasting, both TimelyGPT and all transformer baselines were configured with 18 million parameters. This number is in line with the scaling law outlined in Section 4.1. Model parameters and architectures for other tasks are provided in Appendix C.5.
>
>
> **3.b. Are the baselines pre-trained? How do you pre-train baselines?**
>
> R3.b: For fair comparison, all transformer baselines (PatchTST, Informer, Autoformer, Fedformer, TST, CRT) are pre-trained.
>
> The pre-training methods of TST and CRT are detailed in the Introduction section. PatchTST employs TST's masking pre-training approach, with 40\% of the patches masked. Other transformers also use a similar masking strategy, masking 40\% of timesteps following TST's methodology. The details of pre-training and fine-tuning can be found in Appendix. C.4.

---

> > ### Comment · Reviewer_5JkM · 2023-11-22
> > **Thank you for your detailed response.**
> >
> > The added experiments are very helpful. My concerns on the experiments have been well addressed.
> >
> > However, the components of this works are quite similar to those previously proposed works in other domains, which is also discussed by other reviewers like reviewer ZDVD. The designs are lack of motivations and may need additional explanations of the advantages on ultra long forecasting.
> >
> > Thus, I raise my score to 6.

---

> > > ### Author Response · Authors · 2023-11-22
> > > **Response to Reviewer 5JkM (new Q)**
> > >
> > > Thanks for your constructive feedback. There are ***three points*** we would like to make.
> > >
> > > **new Q: However, the components of this works are quite similar to those previously proposed works in other domains, which is also discussed by other reviewers like reviewer ZDVD (3 points). The designs are lack of motivations and may need additional explanations of the advantages on ultra long forecasting.**
> > >
> > > new R: For the concern about ***component utilization in other domains***, it is true that the components in TimelyGPT (like xPOS and RetNet) were originated   in NLP domain. However, their application within time-series transformers remains largely unexplored. Existing time-series transformers (e.g., Informer, Autoformer, Fedformer, PatchTST) rely on absolute position embedding, limiting their ability of extracting temporal information [1] and of performing extrapolation [2]. Our TimelyGPT uniquely leverages xPOS to effectively capture long-term and short-term temporal patterns [3] and to extrapolate to longer sequence of forecasting during inference [4]. Additionally, we also explore the application of the RNN-based transformer on the time-series, combining the inherent sequential modeling strengths of RNNs [5] with transformer scalability. Therefore, our study initiates a shift of focus toward unexplored areas like extrapolable embedding and RNN-based transformers in time-series domains.
> > >
> > > For the concern on our ***design motivations***, we designed TimelyGPT with three key components (xPOS, retention, and convolution), that are specifically tailored to address four time-series properties: (1) trend patterns; (2) sequential nature; (3) multi-scale signals (i.e., patterns revealed at sub-sequence of different lengths); (4) global and local features.
> > > For xPOS, we examine its potential for time-series extrapolation, building on its known capability to model long-term dependencies in text data [4]. We found that xPOS captures long-term **trend patterns** in time-series, facilitating extrapolation over long timesteps.
> > > For the retention module, we draw insights from RNNs, which are recognized for their effectiveness in sequential data modeling, aligning well with the inherent **sequential nature** of time-series data [5].  We use chunk-wise retention to increase the expressiveness of RNN while maintaining computational efficiency to model long seqeunces.
> > > For the convolution module, we seek to use them extract both **local features** [6] and **multi-scale features** [7] from time-series data, representing various characteristics of time-series.
> > > The integration of attention mechanisms and convolution modules effectively capture both **global and local interactions** in time-series data [3].
> > >
> > >
> > > For the comment on the ***advantages on ultra-long forecasting***, we provide an additional explanation with visualization interpretation (Section 4.6). Pre-trained on 4K timesteps, TimelyGPT makes effective inference on 8K timesteps (2K loop-up window as prompt plus 6K forecasting window). This extrapolation ability is consistent with the performance of xPOS-based transformer in NLP domain [4]. For comparison, existing time-series transformers do not exhibit extrapolation ability due to their absolute position embedding [2]. Through  visualizing the inferred trajectories (Fig. 8 and 9), we observe that the modeling of long-term trend patterns drives the extrapolation in the context of time-series. In contrast, the transformer baselines without extrapolation capabilities generate forests flat trajectories that largely deviate from the groundtruth, which aligns with our analysis in Section 2.4 and a recent study [8].
> > >
> > > | Model | Training Length | Inference Length | Extrapolation Ability |
> > > |-------|-----------------|------------------|-----------------------|
> > > | **TimelyGPT** | 4K | 8K | 2X |
> > > | XPOS-Transformer | 1K | 8K | 8X |
> > > | Time-series transformers | 96-512 | 720 | Limited |
> > >
> > > [ref 1] Are transformers effective for time series forecasting?
> > >
> > > [ref 2] Train Short, Test Long: Attention with Linear Biases Enables Input Length Extrapolation
> > >
> > > [ref 3] Conformer: Convolution-augmented Transformer for Speech Recognition
> > >
> > > [ref 4] A Length-Extrapolatable Transformer
> > >
> > > [ref 5] Efficiently modeling long sequences with structured state spaces
> > >
> > > [ref 6] Convolutional Networks for Images, Speech, and Time Series
> > >
> > > [ref 7] Omni-scale CNNs: a simple and effective kernel size configuration for time series classification
> > >
> > > [ref 8] Revisiting Long-term Time Series Forecasting: An Investigation on Linear Mapping.

---

> > > ### Author Response · Authors · 2023-11-23
> > > **Gratitude for the Reviewer 5JkM's Feedback**
> > >
> > > We deeply appreciate your constructive feedback and the time you've invested in reviewing our work.
> > >
> > > We noted that reviewer ZDVD adjusted their rating following our latest revisions. In response, we've diligently addressed additional questions. We would like to know if you might consider reassessing your rating. Your revised evaluation would be greatly appreciated by us.

---

### Official Review · Reviewer_BKAB · 2023-11-02

**Soundness:** 3 good
**Presentation:** 3 good
**Contribution:** 2 fair
**Rating:** 6
**Confidence:** 3

**Summary:**

The paper proposes a Transformer approach for time series modelling. The main components of the proposed architecture are relative positional embeddings that extract both trend and periodic patterns, recurrent attention with time decay and convolutional module that captures local temporal information.

**Strengths:**

The paper is well written and easy to follow. Some of the aspects of the proposed model appear novel although it is hard to pinpoint exactly which parts are novel as most components such as for example xPOS are based on previous work. Strong empirical performance particularly on long range prediction with large data (e.g. Sleep-EDF). Detailed ablation study and empirical evaluation demonstrating the benefits of each added component and the impact of data and model size.

**Weaknesses:**

Authors emphasise the focus on pre-training. However, I could not find results ablating the benefits of pre-training and whether TimelyGPT is particularly well suited for it. Some experimental settings are odd where models are pre-trained on different datasets for classification vs regression. Task specific pre-training deviates from the foundational model paradigm, why was that done?

Figure 3.a results look almost too good to be true, 720 vs 6K prediction window size is a huge difference and one would expect at least some performance decay. At what prediction window size do you start seeing performance decay and what in TimelyGPT makes it so well suited for such long range prediction?

Some components like the temporal convolution module have very specific sets of operators. How sensitive are the results to these choices? I think the paper would also benefit from clearly highlighting novel components vs novel application of previous work.

**Questions:**

Please see the weaknesses section.

---

> ### Author Response · Authors · 2023-11-21
> **Response to Reviewer BKAB (Q1, Q2, Q3)**
>
> We are grateful for your constructive comments. Below we address each question and concern.
>
> **Q1: Authors emphasise the focus on pre-training. However, I could not find results ablating the benefits of pre-training and whether TimelyGPT is particularly well suited for it. Some experimental settings are odd where models are pre-trained on different datasets for classification vs regression. Task specific pre-training deviates from the foundational model paradigm, why was that done?**
>
> R1: The revised version includes a quantitative ablation study on pre-training in Table. 11. The following table shows that TimelyGPT with pre-training obtains a notable increase of 3.89\% in biosignal classification (SleepEDF), while the improvement (0.83\%) is less evident in irregularly-sampled time series (PopHR).
>
> | Method / Classification | Sleep-EDF | PopHR |
> |-------------------------|-----------|-------|
> | **TimelyGPT (with Pre-training)** | **89.21** | **64.89** |
> | TimelyGPT (w/o Pre-training) | 85.32 | 64.06 |
>
> Regarding the question about ``task specific pre-training", we pre-trained TimelyGPT on the same SleepEDF dataset for both forecasting  (Section 4.2) and classification (Section 4.3) downstream tasks. However, the pre-trained model exhibits limited generalization ability across different types of biosignals. For example, model pre-trained on ECG (SleepEDF dataset) does not generalize well on PPG signal. We acknowledge this limitation in Section 5 and identify it as an avenue for future exploration.
>
> **Q2: Figure 3.a results look almost too good to be true, 720 vs 6K prediction window size is a huge difference and one would expect at least some performance decay. At what prediction window size do you start seeing performance decay and what in TimelyGPT makes it so well suited for such long range prediction?**
>
> R2: The ultra-long-term forecasting in Fig 3 shows our most exciting results. The inference length of 8K (2K loop-up window as prompt plus 6K forecasting window) is the maximum length without observing performance decay.  This aligns with the performance of xPOS-based transformer in NLP domain [1]. In the revised version, our visualization results (Section 4.6) explores the underlying mechanism behind xPOS's extrapolation, which is driven by its modeling of trend patterns in time-series.
>
> For comparison, existing time-series transformers (Informer, Autoformer, Fedformer, PatchTST) with absolute position embedding do not exhibit extrapolation ability [2]. In visualization results (Section 4.6), these transformer baselines without extrapolation capabilities generate forests flat trajectories that largely deviate from the groundtruth, which aligns with our analysis in Section 2.4 and a recent study [3].
>
> | Model | Training Length | Inference Length | Extrapolation Ability |
> |-------|-----------------|------------------|-----------------------|
> | **TimelyGPT** | 4K | 8K | 2X |
> | XPOS-Transformer | 1K | 8K | 8X |
> | Time-series transformers | 96-512 | 720 | Limited |
>
>
> [ref 1] A Length-Extrapolatable Transformer
>
> [ref 2] Train Short, Test Long: Attention with Linear Biases Enables Input Length Extrapolation
>
> [ref 3] Revisiting Long-term Time Series Forecasting: An Investigation on Linear Mapping.
>
> **Q3: Some components like the temporal convolution module have very specific sets of operators. How sensitive are the results to these choices? I think the paper would also benefit from clearly highlighting novel components vs novel application of previous work.**
>
> R3: In the revised version, Table 12 compares various convolutional operator configurations on the downstream classification for SleepEDF dataset. Our Temporal Convolution module outperforms two commonly used operators designed for modeling global-local interaction in transformers.
>
> | Configuration | Accuracy (%) |
> |---------------|--------------|
> | **Depth-wise Conv + Point-wise Conv (Ours)** | 89.21 |
> | Point-wise Conv + Depth-wise Conv + Point-wise Conv ([1]) | 88.75 |
> | Depth-wise Conv ([2]) | 88.12 |
> | Point-wise Conv | 87.56 |
> | w/o Temporal Convolution | 87.03 |
>
> We agree that while the proposed architecture of our TimelyGPT is novel, there is a need to explicitly highlight the level of our contribution, be it at the level of the sum of those parts or at the level of the parts themselves. Following your suggestions, we provide an explicit distinction between novel applications (i.e., applying xPOS to time-series data and retention on continuous time-series data) and novel components (i.e., extension of retention for irregular time-series and the temporal convolution module).
>
>
>
> [ref 1] Conformer: Convolution-augmented Transformer for Speech Recognition
>
> [ref 2] Lite Transformer with Long-Short Range Attention

---

> ### Author Response · Authors · 2023-11-23
> **Final Confirmation with Reviewer BKAB**
>
> Thank you for spending your time and effort to review our paper. We greatly appreciate your valuable suggestions and have made **revisions** accordingly.
>
> We are encouraged by your positive assessment for our work, and we would like to ensure that all your concerns have been adequately addressed. If you believe that your concerns have been resolved, we would be grateful if you could consider revising your rating. We highly value your feedback and sincerely appreciate your consideration in this matter.

---

### Official Review · Reviewer_ZDVD · 2023-11-03

**Soundness:** 2 fair
**Presentation:** 1 poor
**Contribution:** 2 fair
**Rating:** 5
**Confidence:** 4

**Summary:**

This paper introduces TimelyGPT, a pre-trained model designed specifically for time series. This model integrates recurrent attention and temporal convolution modules to capture dependencies in long sequences. The recurrent attention leverages the time decay mechanism to handle continuous and irregular observations. Besides, a relative position embedding is introduced to the transformer to help extract dependencies.

**Strengths:**

The summary provided in Table 1, which compares related baselines in terms of the number of parameters and dataset size, is informative and useful for gaining a quick understanding of the landscape of existing approaches.

**Weaknesses:**

- As mentioned in Section 1, it is suggested that *"challenges observed in small-scale time series might stem from overfitting limited data rather than inherent flaws in transformers"*. This statement is a bit confusing as the title suggests that the model can handle long time series. However, it seems that the goal is to address the overfitting to limited data, which implies that only short-time series are available. This appears to be contradictory. Please clarify what problems in time series applications are mainly addressed here.

- Table 1 can be further improved. For example, it is unclear what is meant by 'data size.' Is it the number of time points or the number of windows? And what are the numbers of the proposed model?

- The writing could be further improved. The formulas in section 3.2 are difficult to follow, and it would be helpful to introduce each formula one by one in a logical manner.

- The proposed TimelyGPT refers to the Timely Generative *Pre-trained* Transformer. However, details of the *pretraining* and fine-tuning processes are missing. Specifically, what datasets and how many epochs were used for pretraining? What fine-tuning strategies were used for downstream tasks?

- There are many related works (e.g., TS2Vec, TimesNet) on time series self-supervised representation learning, but there is a lack of systematic investigation, discussion, and comparison.

- Regarding the prediction results in Figure 3, it should be noted that the dataset is not commonly used for forecasting.
Additionally, a too-long prediction horizon may not make sense as real-world prediction scenarios are usually dynamic and unpredictable.

- It is recommended to summarize the prediction results in a table rather than an overly simplified figure. This makes it easier for other works to follow.

- Visualization results are encouraged to be included.

- The use of a relative position embedding is not a new idea and it has been studied in different communities, e.g., transformer variants and time series applications. For example,
> [ref] STTRE: A Spatio-Temporal Transformer with Relative Embeddings for Multivariate Time Series Forecasting
[ref] Improve transformer models with better relative position embeddings

- If the proposed recurrent attention can handle irregular time series well, it would be beneficial to compare it with popular irregular time series methods.
> [ref] Multi-time attention networks for irregularly sampled time series

- The source code is incomplete.

**Questions:**

Please see the Weaknesses.

---

> ### Author Response · Authors · 2023-11-21
> **Response to Reviewer ZDVD (Q1, Q2, Q3, Q4)**
>
> We appreciate the reviewer’s comments. We update our paper and address your concerns here:
>
> **Q1: As mentioned in Section 1, it is suggested that ``challenges observed in small-scale time series might stem from overfitting limited data rather than inherent flaws in transformers". This statement is a bit confusing as the title suggests that the model can handle long time series. However, it seems that the goal is to address the overfitting to limited data, which implies that only short-time series are available. This appears to be contradictory. Please clarify what problems in time series applications are mainly addressed here。**
>
> R1: The focus of our research is on leveraging large-scale datasets (1.2B timesteps) for transformer pre-training, enabling long-sequence time-series representation (4K timesteps). While large time-series datasets have been available, previous time-series transformers have not effectively utilized them, often focusing on modeling small data (like ETT with 69K timesteps) and short-sequence (96 timesteps).
>
> The rationale for our research is twofold. First, in Section 2.1, we point out that the existing transformer-based frameworks are overparameterized for small datasets. Secondly, in Section 4.1, we show that these transformer-based models follow the scaling law and do not perform well when the data are small. To address these limitations, we pre-train TimelyGPT on the large Sleep-EDF dataset, addressing the overfitting in small data scenarios. With large datasets, TimelyGPT can learn long-sequence representation in time-series domain for the first time.
>
> To improve the clarity, we revised that sentence as follows: *We argue that the seemingly inadequacy of the current transformer-based models in modeling time-series data is due to their inability to model long and big time-series data. Once these challenges are resolved, we would observe a typical scaling law comparable to the one observed in the NLP applications.*
>
> **Q2: Table 1 can be further improved. For example, it is unclear what is meant by `data size.' Is it the number of time points or the number of windows? And what are the numbers of the proposed model?**
>
> R2: In the revised Table 1, we now clearly specify \``Dataset size (Timesteps)" (which is analogous to the ``number of tokens" in NLP applications) to clarify data size.
>
> Detailed information about TimelyGPT model parameters and architectures is provided in Appendix. C.5. The following table provides a summary:
>
> | TimelyGPT Setup | Sleep-EDF | PTB-XL + PPGDalia | PopHR |
> |-----------------|-----------|-------------------|-------|
> | **Experiments** | Forecasting & Classification | Classification & Regression | Classification |
> | **Data Size (timesteps)** | 1.2B | 109.2M + 16.6M | 8.4M |
> | **Model Parameters** | 18M | 7.5M | 3M |
> | **Decoder Layers** | 12 | 8 | 8 |
> | **Heads** | 8 | 8 | 4 |
> | **Dim (Queries)** | 320 | 240 | 144 |
> | **Dim (Keys)** | 320 | 240 | 144 |
> | **Dim (Values)** | 640 | 480 | 288 |
> | **Dim (FF Layers)** | 640 | 480 | 288 |
>
> **Q3: The writing could be further improved. The formulas in section 3.2 are difficult to follow, and it would be helpful to introduce each formula one by one in a logical manner.**
>
> R3: Following your recommendation, we have revised Section 3.2 by presenting equation one by one with improved descriptions. Recognizing that Retention is key contribution of RetNet paper, the main text only includes essential equations and descriptions for its understanding. Comprehensive descriptions are in Appendix B.4.
>
> **Q4: The proposed TimelyGPT refers to the Timely Generative Pre-trained Transformer. However, details of the pretraining and fine-tuning processes are missing. Specifically, what datasets and how many epochs were used for pretraining? What fine-tuning strategies were used for downstream tasks?**
>
> R4: In the revised version, we provide details about pre-training and fine-tuning in Appendix. C.4. In the pre-training, TimelyGPT performs next-token prediction using cross-entropy loss. Given a sequence starts with a [SOS] token, each token's output embedding is used to predict next token using right shift. Following PatchTST's finding that end-to-end fine-tuning yields best results [1], we adopt it on the pre-trained TimelyGPT. The pre-training and fine-tuning of TimelyGPT involve 20 and 5 epochs, respectively, with early stopping.
>
> We already provided the details of datasets in Appendix C.2.
>
> [ref 1] A Time Series is Worth 64 Words: Long-term Forecasting with Transformers

---

> ### Author Response · Authors · 2023-11-21
> **Response to Reviewer ZDVD (Q5, Q6, Q7, Q8, Q9)**
>
> **Q5: There are many related works (e.g., TS2Vec, TimesNet) on time series self-supervised representation learning, but there is a lack of systematic investigation, discussion, and comparison.**
>
> R5: In the revised version, we discuss TS2Vec and TimesNet in section of Related Works and evaluate on various downstream tasks. The following tables shows that TimelyGPT surpasses both TS2Vec and TimesNet among these tasks.
>
> | Models | TimelyGPT | TS2Vec | TimesNet |
> |--------|-----------|--------|----------|
> | **Ultra-long-term Forecasting (MSE)** | | | |
> | Window Size 720 | 0.542 | 0.602 | 0.471 |
> | Window Size 2000 | 0.567 | 1.231 | 0.742 |
> | Window Size 6000 | 0.575 | 1.204 | 0.865 |
> | **Classification (Accuracy %)** | | | |
> | Sleep-EDF | 89.21 | 86.21 | 83.58 |
> | Epilepsy | 92.79 | 88.27 | 85.96 |
> | PTB-XL | 86.52 | 82.65 | 79.38 |
> | EMG | 95.87 | 93.77 | 89.26 |
> | **Regression (MAE)** | | | |
> | IEEEPPG | 26.17 | 27.89 | 29.95 |
> | RR | 2.78 | 3.53 | 4.19 |
> | HR | 8.53 | 11.56 | 13.65 |
> | SpO2 | 4.26 | 4.60 | 4.83 |
>
> **Q6: Regarding the prediction results in Figure 3, it should be noted that the dataset is not commonly used for forecasting. Additionally, a too-long prediction horizon may not make sense as real-world prediction scenarios are usually dynamic and unpredictable.**
>
> R6: In Section 2.1, we point out that commonly used forecasting datasets are small data (up to 69K). As demonstrated in Section 4.1, time-series transformers follow scaling law and underperform on small data. Consequently, we focus on large-scale pre-training (up to 1.2B) instead.
>
> Ultra-long-term forecasting is crucial for healthcare applications such as patient health monitoring. Biosignals like Sleep-EDF are sampled at a high rate of 100 HZ.  Forecasting 6000 timesteps helps doctors effectively monitor patients' health status in the next 1 minute. Short-term forecasting models fall short in real-world healthcare contexts.
>
> **Q7: It is recommended to summarize the prediction results in a table rather than an overly simplified figure. This makes it easier for other works to follow.**
>
> R7: In the revised version, we summarize the forecasting results in Table 9.
>
> **Q8: Visualization results are encouraged to be included.**
>
> R8: In the revised version, we provide visualization result in Section 4.6. Specifically, we show TimelyGPT effectively extrapolates to long-sequences. PatchTST and GPT-2 with absolute embedding rely on linear mapping for forecasting, which aligns with our analysis in Section 2.4 and a recent study [1]. Our visualization results indicate benefits of integrating extrapolable embeddings into time-series transformers for future research.
>
> [ref 1] Revisiting Long-term Time Series Forecasting: An Investigation on Linear Mapping
>
> **Q9: The use of a relative position embedding is not a new idea and it has been studied in different communities, e.g., transformer variants and time series applications. For example,**
>
> **[ref] STTRE: A Spatio-Temporal Transformer with Relative Embeddings for Multivariate Time Series Forecasting**
>
> **[ref] Improve transformer models with better relative position embeddings**
>
> R9: In the revised version, we have referenced STTRE in Section 2.3. However, the STTRE [1] was available online on 09/30/2023. According to the ICLR Reviewer Guide, there's no requirement to compare with papers published "on or after May 28, 2023". Moreover, STTRE is spatio-temporal transformer and another paper [2] addresses NLP task only. Although relative position embeddings are studied in various domains, TimelyGPT is the first work to thoroughly investigates its advantages in time-series applications. TimelyGPT is also the first work to explore the underlying mechanism of extrapolable embedding in time-series modeling. It opens new avenues for future research to transition from absolute to relative position embeddings in time-series domain.
>
> [ref 1] STTRE: A Spatio-Temporal Transformer with Relative Embeddings for Multivariate Time Series Forecasting
>
> [ref 2] Improve transformer models with better relative position embeddings

---

> ### Author Response · Authors · 2023-11-21
> **Response to Reviewer ZDVD (Q10, Q11)**
>
> **Q10: If the proposed recurrent attention can handle irregular time series well, it would be beneficial to compare it with popular irregular time series methods.**
>
> **[ref] Multi-time attention networks for irregularly sampled time series**
>
> R10: In the revised version, we compared TimelyGPT with the recommended baseline (mTAND) [1], along with three widely-recognized baselines (GRU-D [2], SeFT [3], RAINDROP [4]) for irreguarly-sampled time-series. The full results can be found in Table 10.
>
> The following table shows that TimelyGPT exhibits superior performance (average AUPRC of 57.6\%), surpassing the SOTA algorithm mTAND (56.4\%). It also exceeds other baselines, including GRU-D (55.8\%), SeFT (48.7\%) and RAINDROP (54.6\%). TimelyGPT secures the top rank (1.50) among baseline models.
>
> | Phenotype (AUPRC) | TimelyGPT | mTAND | GRU-D | SeFT | RAINDROP |
> |-------------------|-----------|-------|-------|------|----------|
> | AMI               | **0.612** | 0.607 | 0.584 | 0.524 | 0.557 |
> | Asthma            | 0.568 | 0.557 | 0.560 | 0.479 | **0.579** |
> | CHF               | **0.637** | 0.616 | 0.599 | 0.548 | 0.582 |
> | COPD              | 0.541 | **0.547** | 0.533 | 0.452 | 0.516 |
> | Diabetes          | **0.563** | 0.553 | 0.495 | 0.475 | 0.528 |
> | HTN               | 0.586 | 0.571 | **0.599** | 0.498 | 0.531 |
> | Schizo            | 0.551 | **0.561** | 0.513 | 0.463 | 0.526 |
> | IHD               | 0.584 | 0.574 | **0.586** | 0.496 | 0.549 |
> | HIV               | 0.549 | **0.558** | 0.547 | 0.460 | 0.524 |
> | Epilepsy          | **0.560** | 0.539 | 0.552 | 0.472 | 0.525 |
> | Autism            | **0.576** | 0.544 | 0.558 | 0.487 | 0.560 |
> | ADHD              | **0.583** | 0.542 | 0.565 | 0.494 | 0.578 |
> | **Average**       | **0.576** | 0.564 | 0.558 | 0.487 | 0.546 |
> | **Average Rank**  | **1.50** | 2.50 | 2.83 | 5.00 | 3.25 |
>
> [ref 1] Multi-time attention networks for irregularly sampled time series.
>
> [ref 2] Recurrent neural networks for multivariate time series with missing values
>
> [ref 3] Set Functions for Time Series
>
> [ref 4] Graph-Guided Network for Irregularly Sampled Multivariate Time Series
>
> **Q11: The source code is incomplete.**
>
> R11: We already provided the source code of TimelyGPT for reproduction purpose. The organized code will be available on Github after acceptance.

---

> ### Author Response · Authors · 2023-11-23
> **Final Confirmation with Reviewer ZDVD**
>
> Thank you for spending your time and effort to review our paper. We greatly appreciate your valuable suggestions and have made **revisions** accordingly.
>
> We understand that you initially had reservations about our work, and we would like to ensure that all your concerns have been adequately addressed. If you believe that your concerns have been resolved, we would be grateful if you could consider revising your rating. We highly value your feedback and sincerely appreciate your consideration in this matter.

---

> ### Comment · Reviewer_ZDVD · 2023-11-23
>
> I appreciate the authors' detailed responses and I'm satisfied with the supplementary experiments on irregular time series. However, Their answers just partially addressed my concerns.  I have decided to upgrade my rating to 5.
>
> There are still some aspects of the authors' responses that are not satisfactory. Specifically:
>
> - The writing regarding the formulation of the methods still needs further improvement. For instance, there is a lack of shape information for many tensor variables.
> - The discussion on self-supervised representation learning is insufficient. In the revised version, the modification "Additionally, self-supervised representation learning techniques in time series, such as TS2Vec and TimesNet, offer valuable representation learning capabilities for forecasting tasks (Yue et al., 2022; Wu et al., 2023)" appears to be oversimplified.
> - Including more commonly used datasets, such as electricity and traffic, may help demonstrate the model's performance on general datasets. However, this suggestion applies only if the authors have not explicitly limited the scope of their research to healthcare contexts in the title.
> - Table 6 is lacking information on the training, validation, and test sets. Besides, it would be better to add links to these datasets if they are publicly accessible.
> - The prediction results in Figure 9 are remarkably good. The time series trends seem to be very challenging. I would like to see the reasons why the model can model such difficult trends. It would be beneficial to visualize the historical data as well to see if the model relies on similar historical knowledge to accurately predict such challenging trends.
> - Regarding predictive transformers like Autoformer, it is unclear how they can be applied to classification tasks.
> - Another crucial issue is the out-of-memory error often encountered when using Autoformer for long sequence processing. How was this addressed in your long sequence prediction experiments?
> - Transformers often face high complexity and struggle to handle long historical sequences, thereby limiting their performance. In NLP, there are various improved variants of attention mechanisms that can be applied to very long sequential data. I did not see any specific comparisons of these variants in the context of long-range prediction tasks.
>
> Due to time constraints, I would like to have a brief discussion with the authors regarding a specific question:
>
> - There is a perspective here that overparameterized deep learning networks do not overfit [1]. What is your viewpoint on this?
>
> [1] Zhou Z H. *Why over-parameterization of deep neural networks does not overfit?*. Science China Information Sciences, 2021, 64: 1-3.

---

> ### Author Response · Authors · 2023-11-23
> **Response to Reviewer ZDVD (New Q1-Q5)**
>
> Thanks for your constructive feedback and insightful suggestion. Due to constrained time, we would like to address your main concerns as below.
>
> **Q1: The writing regarding the formulation of the methods still needs further improvement. For instance, there is a lack of shape information for many tensor variables.**
>
> R1: This suggestion provides valuable assistance in improving the readability for our work. In response to your suggestion, we have revised our paper and clearly stated the shape of every variable in Section 3.1 and Section 3.2.
>
> **Q2: The discussion on self-supervised representation learning is insufficient. In the revised version, the modification "Additionally, self-supervised representation learning techniques in time series, such as TS2Vec and TimesNet, offer valuable representation learning capabilities for forecasting tasks (Yue et al., 2022; Wu et al., 2023)" appears to be oversimplified.**
>
> R2: In the revised version, we include an analysis for the challenge faced by TimesNet. Given TimesNet's dependence on frequency-domain information and Fourier transform, the masking in time-series leads to the distribution shift [1], adversely impacting the performance.
>
> [ref 1] Self-Supervised Time Series Representation Learning via Cross Reconstruction Transformer
>
> **Q3: Including more commonly used datasets, such as electricity and traffic, may help demonstrate the model's performance on general datasets. However, this suggestion applies only if the authors have not explicitly limited the scope of their research to healthcare contexts in the title.**
>
> R3: In our research, we mainly focus on time-series in healthcare contexts. We highlight at the beginning of our paper: ``Time-series data mining holds significant importance in healthcare, given its potential to trace patient health trajectories and predict medical outcomes (Ma et al., 2023b; Eldele et al., 2021; Fawaz et al., 2019). In the field of healthcare, there are two primary categories: continuous and irregularly-sampled time-series data".
>
> **Q4: Table 6 is lacking information on the training, validation, and test sets. Besides, it would be better to add links to these datasets if they are publicly accessible.**
>
> R4: Table 6 in Appendix C.1 (Dataset Description) organizes the information regarding datasets, where all biosignal datasets we cited in Appendix C.1 are publicly available. We have included the pre-processing details in Appendix C.2 (Data Pre-processing). We summarize the train/validation/test sets as follows:
>
> | Method | Task | Train | Validation | Test |
> |----------|------|-------|------------|------|
> | Sleep-EDF | Forecasting | 80% | 10% | 10% |
> | Sleep-EDF, Epilepsy, PTB-XL, EMG  | Classification | 85% | 5% | 10% |
> | PPG-Dalia, IEEEPPG, RR, HR, Sp02 | Regression| 85% | 5% | 10%|
> | PopHR | Classification | 80% | 10% | 10% |
>
> **Q5: The prediction results in Figure 9 are remarkably good. The time series trends seem to be very challenging. I would like to see the reasons why the model can model such difficult trends. It would be beneficial to visualize the historical data as well to see if the model relies on similar historical knowledge to accurately predict such challenging trends.**
>
> R5: The effectiveness of trend modeling in Fig.8 is largely attributed to the xPOS embedding in our TimelyGPT framework. This position embedding utilizes exponential decay $\gamma$ to attenuate the influence of distant tokens, aiding in capturing long-term dependency and extrapolation ability in NLP applications [1]. In the time-series analysis, we found that the exponential decay can effectively capture long-term trend patterns (Fig. 9), which aligns with Fig. 2 in xPOS paper [1].
>
> | Model | Training Length | Inference Length | Extrapolation Ability |
> |-------|-----------------|------------------|-----------------------|
> | **TimelyGPT** | 4K | 8K | 2X |
> | XPOS-Transformer | 1K | 8K | 8X |
> | Time-series transformers | 96-512 | 720 | Limited |
>
> In the revised version, we include visualization results of a 8K-timestep history, a 2k look-up window, and a 6K forecasting window for a sleep stage transition  (Fig. 10). For the temperature signal, we observe distinct historical patterns that differ from the forecasting window, highlighting the temperature changes over 16K timesteps (160 seconds). Leverage pre-training power, Section 4.6 found that ``the small bump in the prompt right before the 1000-th timestep is a typical indicator for temperature drop". In contrast, the periodic EEG signal shows consistent periodic patterns throughout the 16K timesteps.
>
> [ref 1] A Length-Extrapolatable Transformer

---

> ### Author Response · Authors · 2023-11-23
> **Response to Reviewer ZDVD (New Q6-Q9)**
>
> **Q6: Regarding predictive transformers like Autoformer, it is unclear how they can be applied to classification tasks.**
>
> R6: Transformers designed for forecasting (like Autoformer, Fedformer, and PatchTST) typically do not incorporate classification tasks directly in implementation. Nonetheless, models like TST and CRT perform classification tasks by using a linear layer with averaging last layer's embeddings.
>
> **Q7: Another crucial issue is the out-of-memory error often encountered when using Autoformer for long sequence processing. How was this addressed in your long sequence prediction experiments?**
>
> R7: To address memory limitations often occurred in the training of transformers, we conducted our experiments on a A-100 GPU with 40GB memory. Additionally, we leveraged multiple strategies to optimize memory usage in our submitted code: (1) gradient checkpointing; (2) gradient accumulation; (3) PyTorch's garbage collection.
>
> **Q8: Transformers often face high complexity and struggle to handle long historical sequences, thereby limiting their performance. In NLP, there are various improved variants of attention mechanisms that can be applied to very long sequential data. I did not see any specific comparisons of these variants in the context of long-range prediction tasks.**
>
> R8: While comparing various long-context attention modules would be valuable, the Retnet paper has already conducted such comparisons, including RWKV, H3, Heyna, and Linear Transformer. Applying all these models specifically to time-series data would be an extensive work beyond our current scope.
>
> **Q9: Due to time constraints, I would like to have a brief discussion with the authors regarding a specific question:**
>
> **There is a perspective here that overparameterized deep learning networks do not overfit [1]. What is your viewpoint on this?**
>
> **[1] Zhou Z H. Why over-parameterization of deep neural networks does not overfit?. Science China Information Sciences, 2021, 64: 1-3.**
>
> R9: This paper [1] provides a valuable perspective on overparameterization in DNN, specifically in the context of conventional machine learning theory. By viewing a MLP as a combination of large-size feature space transformation and small-size classifier construction, it is possible for the number of parameters to surpass the sample size without overfitting.
>
> However, in the context of Transformer, it typically follows the Neural Scaling Law [2]. The study [2] observed that ``For a smaller fixed D (dataset size), performance stops improving as $N$ (model size) increases and the model begins to overfit". It shows that both model parameters and dataset size need to be scaled up simultaneously to improve model performance. Our finding in Fig. 2, along with studies in other domains [3], corroborate this scaling law. Thus, while the paper [1] provides valuable insights, it may not fully align with the scaling behaviors exhibited by Transformer models.
>
> [ref 1] Why over-parameterization of deep neural networks does not overfit?
>
> [ref 2] Scaling Laws for Neural Language Models
>
> [ref 3] Scaling Vision Transformers

---

### Meta-Review · Area_Chair_pw56 · 2023-12-05

**Metareview:**

This paper proposes TimelyGPT, a novel transformer architecture designed for long-time series representation and modeling. TimelyGPT integrates three key components: xPOS embeddings for segmenting long sequences and enabling extrapolation, a recurrent attention retention mechanism for modeling global context, and convolution modules for capturing local interactions. The paper highlights the originality and significance of TimelyGPT, emphasizing its innovative techniques like future representation decoding and extrapolation. It also acknowledges the importance of scaling analysis, diverse pre-training data, and theoretical grounding for further research. Overall, TimelyGPT shows promising results on both biosignal and irregular time series data, paving the way for future advancements in long-time series modeling. However, further work on scalability, diverse pre-training data, and theoretical analysis are needed to pass the acceptance bar. Overall, the concerns dominate the rejection decision.

**Justification For Why Not Higher Score:**

- Limited analysis of computational overhead and scaling behavior compared to other transformers.
- Pre-training experiments only used one large-scale time series dataset - more real-world pretraining scenarios are needed
- Classification task comparisons limited in dataset and architecture range.
- Limited theoretical analysis or frameworks relating components to time series properties.
- No hyperparameter sensitivity analysis to provide insights into optimal configurations.
- Limited analysis on new task few-shot generalization
- Limited analysis on the architecture

**Justification For Why Not Lower Score:**

- Novel and creative integration of relative position modeling, recurrent attention, and convolutions.
- Significant contribution by applying these architectures specifically for long-time series modeling.
- Innovative idea of decoding future representations and performing extrapolation.
- Clear structure with logical flow of techniques, experiments, results, and analyses.
- Clear writing explaining intuition and technical details.

---

### Decision · Program_Chairs · 2024-01-16

Reject